# GISTIFY! CODEBASE-LEVEL UNDERSTANDING VIA RUNTIME EXECUTION

**Hyunji Lee**[*1]    **Minseon Kim**[2]    **Chinmay Singh**[2]    **Matheus Pereira**[2]    **Atharv Sonwane**[3]

**Isadora White**[4]    **Elias Stengel-Eskin**[5]    **Mohit Bansal**[1]    **Zhengyan Shi**[2]

**Alessandro Sordoni**[2]    **Marc-Alexandre Côté**[2]    **Xingdi Yuan**[2]    **Lucas Caccia**[*2]

[1]University of North Carolina at Chapel Hill    [2]Microsoft Research
[3]Cornell University    [4]University of California San Diego    [5]University of Texas at Austin

hyunjil@cs.unc.edu    debug-gym@microsoft.com

## ABSTRACT

As coding agents are increasingly deployed in large codebases, the need to automatically design challenging, codebase-level evaluation is central. We propose GISTIFY, a task where a coding LLM must create a single, minimal, self-contained file that can reproduce a specific functionality of a codebase. The coding LLM is given full access to a codebase along with a specific entrypoint (e.g., a python command), and the generated file must replicate the output of the same command ran under the full codebase, while containing only the essential components necessary to execute the provided command. Success on GISTIFY requires both structural understanding of the codebase, accurate modeling of its execution flow as well as the ability to produce potentially large code patches. Our findings show that current state-of-the-art models struggle to reliably solve GISTIFY tasks, especially ones with long executions traces[1].

## 1 INTRODUCTION

Large language models (LLMs) are increasingly being used in code-related tasks, powering applications in debugging (Yuan et al., 2025) and agentic code generation (Yang et al., 2024; Liang et al., 2025). Thus, the ability to handle isolated snippets and reasoning across entire codebases, including complex file and module relationships, is becoming increasingly essential. Yet, the evaluation toolkit for assessing such capabilities has lagged behind. Recent evidence shows that widely-adopted repository-level benchmarks such as SWE-bench (Jimenez et al., 2024) and RepoBench (Liu et al., 2023b) still do not *require full* reasoning over the whole execution and could be solved through heuristic shortcuts or retrieval of localized patches (Aleithan et al., 2024; Liang et al., 2025). Moreover, because many of these datasets rely on GitHub issues or pull requests for construction, they are not easily generalizable to arbitrary repositories. At the same time, coding agents are increasingly deployed in large, real-world codebases, highlighting the need for *automatically constructed*, broadly applicable, and more challenging repository-level evaluation.

To fill this gap, we introduce the GISTIFY task, which is deliberately inspired by a common practice of how developers navigate and understand unfamiliar repositories. Rather than reading files in isolation, they start from a concrete execution point such as test command or entry script often mentioned in READMEs. Then, they iteratively reason over the runtime behavior such as identifying dependencies, following control paths to uncover the codebase's structure and functionality. GISTIFY formalizes this practice by requiring an (agentic) coding model to extract the *gist* of a given command, i.e. to generate a single, self-contained, minimal, and executable gistified file that faithfully reproduces the runtime behavior of a given command as when using the original full codebase (Figure 1). In addition to serving as a challenging coding task, such gistified repositories might give

---

[*]Denotes equal contribution

[1]Our code is available at https://github.com/microsoft/gistify

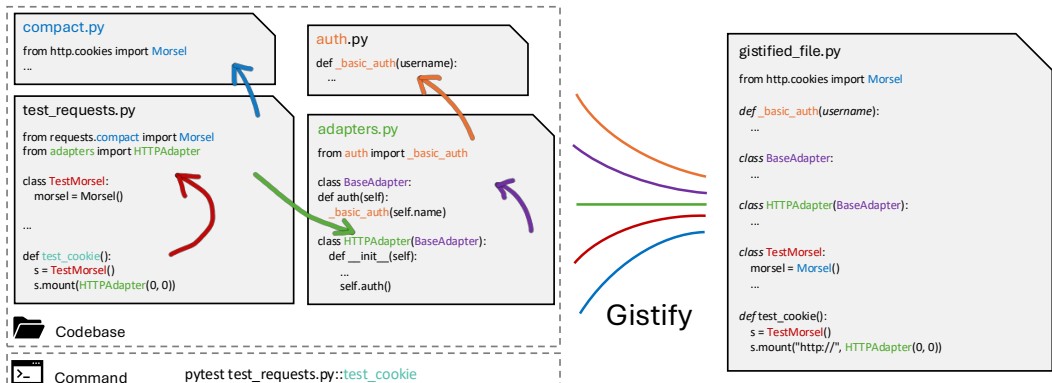

Figure 1: The GISTIFY task: given a codebase and a command of entrypoint, the goal is to generate a minimal, self-contained gistified code file that faithfully reproduces the original runtime behavior using code from the given codebase.

human coders a better understanding of a specific functionality of a given codebase, or even a way to *export* the single functionality of interest without inheriting heavy dependencies.

To perform well in GISTIFY, an agent should generate a single gistified file that satisfies *four* key requirements: it should be **self-contained**, including all necessary components from the codebase so that it can be executed independently; it should ensure **execution fidelity**, producing the same outputs as the original codebase under the given command; it should satisfy **minimality**, retaining only the essential code required for execution without redundant or extraneous lines; and it should guarantee **faithful preservation**, avoiding hallucinated or fabricated code and relying solely on content from the original codebase. To assess model performance, we introduce evaluation metrics that align with these requirements, providing a systematic way to measure codebase-level understanding. GISTIFY requires agents to follow the execution path through the codebase without bypassing modules, i.e., understanding how relevant objects are modified along the way, and identifying which classes or functions can be simplified or removed. Since even moderately sized codebases exceed the context window of current LLMs, success also requires effective search capabilities.

The advantages that GISTIFY brings are multiple: first, it provides direct insight into the ability of models to reason at the codebase level with an understanding of runtime execution, rather than on isolated code snippets. Second, it is lightweight and broadly applicable: it requires only the repository and a test suite (or any other collection of entrypoints with a well-defined expected output) and does not require issue logs or pull requests. This allows automatic construction of challenging tasks for arbitrary repositories, including private ones. Finally, gistified files themselves are valuable outputs: by compressing a specific feature of a large codebase into a minimal file, they can be applied to various downstream tasks, including automated debugging or error localization.

We conduct experiments across a variety of frameworks (mini-SWE-agent, SWE-agent, and Copilot) and models (GPT-5-mini, GPT-5, Claude-3.7-Sonnet, and Claude-Sonnet-4) and uncover several interesting findings. First, even widely used, high-performing frameworks and models struggle to create a successful gistified file, especially when execution traces are long and have high coverage on the repositories. Second, faithfully reproducing the test function in the generated file is a strong indicator of gistified performance, as it serves as the starting step for reasoning about execution traces. Third, enabling execution tools yields small but consistent performance gains, and additionally providing global code context and runtime information further boosts performance. Finally, agentic models benefit from dynamically deciding what to read and refine their reasoning through multi-step trajectories, outperforming static approaches.

## 2 RELATED WORKS

### 2.1 CODEBASE-LEVEL UNDERSTANDING BENCHMARK

Previous work has introduced a variety of benchmarks to evaluate LLMs on codebase-level code understanding. These generally fall into three categories: question answering, code synthesis,

and mapping natural language specifications to the entire codebase. Several benchmarks introduce codebase-level question-answering (Strich et al., 2024; Li et al., 2024b; Sahu et al., 2024; Chen et al., 2025; Hu et al., 2024; Fu et al., 2025). In these settings, the model must correctly answer questions that require an understanding of the codebase. The questions are drawn from various sources, including real-world GitHub issues and queries resembling those asked of tools like Copilot. Another line of work evaluates whether models can synthesize code by leveraging information distributed across multiple files in the codebase (Zhang et al., 2023; Liu et al., 2023b; Ding et al., 2023; Li et al., 2024a; Yu et al., 2024). These benchmarks include tasks such as retrieval-augmented completion, cross-file refactoring, and more specialized settings such as sketch-based coding or codebase evolution. Moreover, there is a line of benchmark that maps natural language specifications to entire code repositories, leveraging hierarchical or multi-stage representations to capture inter-file relationships and maintain consistency across a codebase (Tang et al., 2023; Zan et al., 2024; Ni et al., 2025). Our work tackles a more complex setting, where models must reason over *full* execution traces and examine multiple files, making the task challenging, and even widely used agentic models struggle alongside static ones.[2]

## 2.2 RUNTIME EXECUTION

Various works have introduced benchmarks to evaluate LLMs' ability to reason over code execution at runtime (Gu et al., 2024; Chen et al., 2024; Xie et al., 2025; Beger & Dutta, 2025; Hu et al., 2025). These benchmarks typically test whether models can predict execution traces or intermediate states such as variable values, control flow, or data dependencies—given code and inputs, or alternatively, infer inputs from code and outputs. Some benchmarks further extend this paradigm by leveraging execution traces to construct new problems through program composition, thereby varying complexity in a principled way. Beyond evaluation, execution traces have also been incorporated into training pipelines to strengthen models' runtime reasoning abilities (Liu et al., 2023a; Ding et al., 2024). By augmenting pre-training and fine-tuning with execution states, paths, and coverage signals, these methods help models capture program dynamics and generalize to execution-aware tasks. At inference time, several frameworks leverage runtime feedback to iteratively guide models in debugging or completing partial programs, thereby improving performance on execution-driven tasks (Zhong et al., 2024; Xue et al., 2024). In this work, we extend prior approaches by going beyond reasoning over execution traces to also reformulate programs; the model not only tracks execution but also identifies how to compress and organize code into a concise, coherent file. We further show that this capability serves as a useful tool at inference time, helping models better structure and complete execution-driven tasks.

## 3 GISTIFY

### 3.1 TASK DEFINITION

As shown in Figure 1, when given a codebase and a command as input, the coding agent must generate a single gistified file that reproduces the runtime behavior of the original codebase under the given command. Specifically, the gistified file must satisfy the following requirements.

**Self-Contained:** All necessary components from the given codebase must be included so that the gistified file can be executed standalone, i.e. without relying on the codebase. The model must identify all relevant modules and dependencies, demonstrating understanding of inter-file relationships.

**Execution Fidelity:** Executing the gistified file must replicate the original codebase's runtime behavior, ensuring the model captures the dynamic execution, not just static code patterns.

**Minimalism:** Only the code essential to reproducing the runtime behavior should be preserved, with unused functions and objects pruned. This requires fine-grained understanding of the code to identify which lines are actually executed and essential for the task.

**Grounded Preservation:** No hallucinated code may be introduced. All content must be derived directly from the original codebase. This ensures the task evaluates the model's understanding of the codebase, rather than its ability to generate arbitrary code that happens to satisfy the command.

---

[2]See Appendix A.1 for related works regarding "Methods for Codebase-level Understanding"

## 3.2 EVALUATION PROTOCOL

There are two inputs to a GISTIFY task: i) a docker image containing the target codebase, for consistent evaluation; ii) an entrypoint, such as a pytest command on one of the tests in the codebase. Test cases are existing entrypoints one can easily leverage, but broadly, any command that the user would want to use to run a functionality of the existing codebase is allowed.

All models are prompted to generate a gistified file for the entrypoint. We can programmatically verify whether the expected behavior is preserved when the ground-truth test is run within this setup. Here, we focus on comparing outputs of test commands. Once the model generates the gistified file, to ensure that execution for evaluation is based on the original test, we integrate the test code from the original codebase to the gistified file and execute it. This ensures that the model does not cheat by modifying the test.

## 3.3 METRICS

Once a gistified file is generated, we evaluate it using the given execution command. The evaluation considers three dimensions, aligned with the task requirements, to provide a comprehensive measure of a model's ability to reason over an entire codebase and understand its execution behavior. See Appendix B.1 for more details.

**Execution Fidelity** is a binary metric where 1 means the gistified file runs successfully and produces the same output as the original codebase when executed under the given command; otherwise, it is 0. Failures include cases where the file is not runnable or yields different outputs. The comparison checks for tests pass/fail consistency and stdout/stderr matching.

Formally, let $c$ denote the given command, $\mathcal{C}$ a given codebase, and $\mathcal{G}$ a gistified file. Define $\mathrm{runs}(c, \mathcal{C})$ as an indicator of whether $c$ executes without crashing when running over $\mathcal{C}$, and $\mathrm{out}(c, \mathcal{C})$ returns the set of outputs and error traces from running $c$ with $\mathcal{C}$. Then, execution fidelity is defined as

$$\mathbb{1}\big[\mathrm{runs}(c, \mathcal{G}) \wedge \mathrm{out}(c, \mathcal{G}) = \mathrm{out}(c, \mathcal{C})\big], \tag{1}$$

where $\mathbb{1}[\cdot]$ is the indicator function.

**Line Execution Rate** measures minimality by calculating the fraction of lines in the gistified file that are actually executed under the given command. A 100% execution rate means all lines are essential, indicating a focused and concise file. This metric is only computed for files that run successfully, since the execution trace is required to determine which lines are run.

Formally, let $\mathcal{L}_{\mathrm{exec}}(\mathcal{G})$ be a list of executable lines (i.e., no comments) in $\mathcal{G}$. Then, the Line Execution rate is defined as

$$\frac{1}{|L_{\mathrm{exec}}(\mathcal{G})|} \sum_{\ell \in L_{\mathrm{exec}}(\mathcal{G})} \mathbb{1}[\ell \text{ is executed}]. \tag{2}$$

**Line Existence Rate** measures the proportion of code in the gistified file that is directly preserved from the original codebase. Specifically, lines of code are grouped into blocks (classes, functions, or top-level units), and matches are computed block by block while respecting the code hierarchy. This helps avoiding false matches from common lines appearing in unrelated parts of the codebase. To ensure robustness, we normalize across common variations such as indentation, multi-line statements, and imports. A 100% existence rate indicates full fidelity to the original codebase without hallucination.

Formally, let $\mathcal{B}_{\mathcal{G}}$ and $\mathcal{B}_{\mathcal{C}}$ be the sets of blocks in the gistified file and the original codebase, respectively. For a block $b$, let $\mathcal{L}(b)$ represent its set of lines. Then, the existence rate is defined as

$$\frac{1}{\sum_{b \in \mathcal{B}_{\mathcal{G}}} |\mathcal{L}(b)|} \sum_{b \in \mathcal{B}_{\mathcal{G}}} \sum_{\ell \in \mathcal{L}(b)} \mathbb{1}\{\ell \in \mathcal{L}_{\mathcal{C}}(b)\}, \tag{3}$$

where $\mathbb{1}\{\ell \in \mathcal{L}_{\mathcal{C}}(b)\} = 0$, if no matching block exists in $\mathcal{B}_{\mathcal{C}}$.

Table 1: Average Performance over three agentic frameworks with four models. We evaluated over 25 tests over 5 repositories. Execution Fidelity is shown as *w/o exec*, and *w execution tools*. Line Existence and Execution are averaged across the two settings for clarity.

| Framework | Model | Execution Fidelity (wo exec / w. exec) | Line Existence | Line Execution |
|---|---|---|---|---|
| mini-SWE-agent | GPT-5-mini | 17.1 / 24.0 | 44.9 | 61.2 |
| | GPT-5 | 51.0 / 54.0 | 56.8 | **83.1** |
| | Claude-3.7 | 38.7 / 43.3 | 66.0 | 69.2 |
| | Claude-4 | 54.0 / **55.3** | **67.0** | 75.7 |
| SWE-agent | GPT-5-mini | 30.9 / 45.3 | 47.9 | 74.8 |
| | GPT-5 | 30.7 / 46.0 | 48.3 | **81.7** |
| | Claude-3.7 | 40.7 / 46.0 | **66.8** | 69.9 |
| | Claude-4 | 56.7 / **57.3** | 66.3 | 72.9 |
| Copilot | GPT-5-mini | 58.0 / 55.3 | 62.4 | 77.8 |
| | GPT-5 | 58.7 / 60.7 | 66.9 | **81.4** |
| | Claude-3.7 | 43.3 / 56.0 | 63.0 | 74.4 |
| | Claude-4 | 58.7 / **61.3** | **69.6** | 80.3 |

## 4 EXPERIMENTS

### 4.1 SETTING

We conduct experiments using three widely adopted open-sourced frameworks. SWE-Agent (Yang et al., 2024) and GitHub Copilot (Microsoft, 2025) provide a rich scaffolding to LLM-based agents, enabling them autonomously perform software engineering tasks. This includes a set of tools for creating and editing code files, navigating repositories, and executing tests. These frameworks also offer the LLM controllable cache management, and LLMs follow the standard tool-calling format. We also experiment with Mini-SWE-Agent (Yang et al., 2024), a lightweight framework where LLMs only have access to a bash terminal to solve the task. Commands are parsed from the agent output and executed directly. As the task objective is for the model to use *reasoning* over the execution flow rather than ability of tool usage, for the agentic models, we exclude the execution tools ("python", "pytest") in the default setting where execution is disabled. For all three frameworks, unless specified otherwise, hyperparameters and configurations (e.g. system prompts, cache management, tools) are kept to the default values. Please see Appendix

Our evaluation spans four leading LLM variants: GPT-5 (OpenAI, 2025a), GPT-5-mini (OpenAI, 2025b), Claude-3.7-Sonnet (Anthropic, 2025a), and Claude-Sonnet-4 (Anthropic, 2025b), offering different cost / performance tradeoffs. For ease or reading, we will refer to the last two models as Claude-3.7 and Claude-4. We use a 128K token limit for all models. All experiments ran are capped at 50 steps, after which whatever is generated at this moment in the gistifed file is submitted for evaluation.

On the data side, we experiment over with widely used GitHub repositories which are present in SWE-Bench (`requests`, `pylint`, `flask`, `scikit-learn`, `seaborn`). We also explore an additional repository, `debug-gym` (Yuan et al., 2025)[3]. This library is relatively new and importantly does not overlap with SWE-Bench. We extract and filter test sets for each repository. Namely, we remove tests whose execution is dependent on the test's file location. For the main experiment, we evaluate over 25 tests for each of the 5 repositories[4]. More details regarding the evaluation setup and prompt can be found in the Appendix C.

### 4.2 RESULTS

We begin by giving an overview of the main results presented in Table 1. We report results for our main evaluation protocol, where the model does not have access to execution tools (e.g. "python" and "pytest" commands), as well as the alternative. Examples of gistified files are in Appendix D.1.

---

[3]We provide link to all the GitHub repositories used in this work in Table 5.

[4]We include results on a larger evaluation set in Appendix D.6 and observe that they are consistent with those obtained on the smaller subset.

Table 2: Average error rates (%) of different failure reasons when running SWE-agent across models. Error cases are categorized into four groups. The numbers in parentheses indicate the number of errors for each category.

| Models | Import Error | File Creation Failure | Missing Test Function | Pytest Runtime Error |
|---|---|---|---|---|
| GPT-5-mini | 2.1 (2) | 11.3 (11) | 76.3 (72) | 10.3 (10) |
| GPT-5 | 5.2 (4) | 10.4 (8) | 77.9 (60) | 6.5 (5) |
| Claude-Sonnet-3.7 | 20.0 (10) | 20.0 (10) | 2.0 (1) | 58.0 (29) |
| Claude-Sonnet-4 | 32.5 (13) | 10.0 (4) | 7.5 (3) | 50.0 (20) |

**Claude-4 shows the most robust performance**. Across all frameworks and configurations. Claude-4 consistently provides the best performance, reaching a 54-60% average solve rate. Moreover, the model shows the highest values of Line Existence, meaning that it was the most successful model at faithfully extracting code from the original codebase. We note however that GPT-5 produces the most concise outputs, with Line Execution rate markedly higher than other models.

**Frontier models (GPT-5 / Claude-4) are strong bash users**. When looking at performance on mini-swe-agent, where the models only have access to a bash terminal to solve the task, both models perform relatively well, solving over half of the tasks. Importantly, this is not the case for smaller and previous-generation models.

**Execution tools are not a silver bullet.** Overall, when comparing performance with and without execution in Table 1, we note that in most cases we observe only a small performance gain. We expected that current coding LLMs could better leverage execution tools: indeed, using tools specifically for runtime execution analysis, such as a debugger, could significantly help solving a gistify task. However, we are not seeing this behavior emerge, even from frontier models. We observed a sharp decrease in performance for the GPT-5 model when evaluated on SWE-Agent without execution tools. We performed a visual inspection and noticed formatting issues when rewriting the input test function. A detailed discussion can be found in Appendix D.2.

**Small(er) models perform well with scaffolding**. We note that GPT-5-mini's performance varies significantly across different evaluation settings, from 17% in a bash-only setup to 58% when provided with a large inventory of tools from the Copilot framework (see Appendix D.3 for a full list). We note that this performance increase is also reflected in the quality of the generated gist, where we see a notable increase in line existence and line execution.

### 4.3 ERROR ANALYSIS OVER EXECUTION FAILURE

We proceed with an analysis of the underlying failure causes, in order to understand which aspect of the GISTIFY task different models struggle with. Table 2 shows that each model tends to fail for different reasons. See Appendix D.4 for detailed examples of each error case.

**Import Error** occurs when the model incorrectly imports the original codebase (e.g., `import requests`) instead of inlining the required modules into the gistified file. We note that this error occurs even as coding LLMs are explicitly prompted not to import the specific packages in question. Perhaps surprisingly, the best performing model, Claude-4, commits this seemingly innocuous error the most out of all four models.

**File Creation Failure** errors arise when the model fails to generate the gistified file. This can happen in two ways: the model exceeds the maximum step limit, or the model terminates the task without any file being generated.

**Missing Test Function** errors occur when the generated gistified file does not contain the function implementation for the test specified in the given command, or implements the test in a different structure. This can happen when the model strips out the content of the test and executes it outside of the pytest wrapper, under e.g. `if __name__ == __main__:`. Claude models tend to avoid this mistake, while this is the main source of error for GPT-5 models, specifically under the SWE-agent

Table 3: Analysis of the effect of different strategies and tools (global information, execution) on the GISTIFY task. We evaluate SWE-Agent with Claude 4 using 50 test instances from the pylint codebase. Max Steps Reached (%) indicates the percentage of runs that terminated because the maximum step limit was reached.

| Ablation | Type | Execution Fidelity | Line Existence | Line Execution | Max Steps Reached (%) |
|---|---|---|---|---|---|
| Base GISTIFY | | 42.0 | 65.0 | 58.3 | 14.6 |
| Prompted Strategies | Tracing | 48.0 | 75.4 | 62.8 | 0.0 |
| | Reading | 50.0 | 77.6 | 62.6 | 3.9 |
| Global Info (Tool) | RepoGraph | 52.0 | 76.1 | 60.1 | 6.0 |
| | Tracing | 56.0 | 75.1 | 65.1 | 0.0 |
| Execution (Tool) | Bash | 52.0 | 73.1 | 64.2 | 16.0 |
| | Edit And Execute | 56.0 | 74.3 | 64.2 | 10.0 |

framework. Importantly, we observe that this error does not happen at random, but rather alongside other execution errors; we attempted to add the missing test function, and it in most cases the test fails to run, i.e. it results in a runtime error. This aligns with the analysis in the next section, showing a strong correlation between the task's success and the fidelity between the original and the generated tests.

**Pytest Runtime Error** occurs when the execution of the generated file fails, either due to a runtime error or because the gistified output does not match the output from the original codebase. The results indicate this is the most common cause of error for the best performing model, Claude-4.

### 4.4 IMPORTANCE OF FAITHFULLY PRESERVING THE TEST FUNCTION

We observe that models frequently modify the test function, despite being provided with explicit instructions to copy without modification, except for unavoidable adjustments (e.g., removing imports). Again, to ensure consistent evaluation, we replace the test function in the gistified file with the original version before evaluation.

To measure such modifications, we define the *Test $F_1$ Score* as the line-level overlap between the *test code* of the original file and the gistified version. High Test $F_1$ Score indicates that the model has successfully identified and copied the correct test function to the gistified file. We observe a strong correlation between Test $F_1$ Score and execution fidelity (correlation=0.76, p=0.01); test instances with higher $F_1$ scores are substantially more likely to produce a successful gistified file. We hypothesize that this arises because in the GISTIFY task, models often reason backwards from the test file, thereby if the model fails from identifying or copying the test function, the subsequent reasoning process is highly likely to fail.

To better understand the impact of the first step—searching, viewing, and copying the test function—we conduct an ablation study where we remove potential failure at this stage. Specifically, we explicitly provide the correct test function body and signature in the prompt, so the model no longer needs to locate or copy it. This isolates the effect of errors in identifying the test function. In this setting, we observe that Test $F_1$ Score improves highly from the base GISTIFY 68.4 to 85.3, along with execution fidelity (from 42.0% to 60.0%). This suggests that accurately handling the test function is a critical first step to do the GISTIFY task successfully. Detailed results are in Appendix D.5.

## 5 ANALYSIS

In this section, we analyze how different strategies and tools affect performance on the GISTIFY task, identify factors that contribute to its difficulty, and experiment with the use of a static coding LLM to gain a deeper understanding of the task. For all experiments, we evaluate 50 test instances drawn from the pylint codebase, a setting where the model generally exhibited modest performance. We use SWE-Agent paired with Claude-Sonnet-4.

## 5.1 Effect of Various Strategies and Tools

In this section, we analyze how different strategies and sources of information affect model performance. We begin with the simplest approach, modifying the prompt to guide the model (*Prompt-Based Guidance*), and then move to more explicit approaches that rely on additional tools: providing global context (*Global Information via Tools*) or feedback from code execution (*Execution-Based Tools*). Detailed descriptions of prompts and tools, along with examples, are provided in the Appendix E.1.

**Prompt-Based Guidance** We first begin with the simplest approach: modifying the prompt to provide explicit task guidance. We experiment over two settings. In the former, we prompt the model to perform step-by-step reasoning, by first predicting the execution traces and then going over them, adding relevant code snippets along the way (*tracing*). In the latter, a similar approach is used, with explicit instructions on *how* to recursively determine the execution traces: starting from the test, identify the relevant components and read the files where they are defined, and repeat until the end (*reading*). As shown in Table 3, we observe that adding such strategies tends to enhance overall metrics, giving both better execution fidelity and more faithful code extractions, as measured by line existence.

**Global Information via Tools** Building on the above observation, we next assess the effect of explicitly providing global context through external tools, rather than predicting it. We examine two tools: (1) *RepoGraph* (Ouyang et al., 2024), which constructs a graph of the codebase where each node represents a line of code and edges capture connections between lines, enabling graph-based search over the entire codebase; and (2) a *Tracing* tool that exposes gold execution traces obtained from running the given test command. Results in Table 3 show that both tools improve performance, with the *Tracing* tool yielding the largest gains. This finding suggests that access to the global context, especially the gold tracing information, substantially strengthens the model's ability to perform runtime reasoning, as it can easily identify which file to look at.

**Execution-Based Tools** In Section 4.2, we saw that enabling execution tools resulted in small but consistent gains overall. In this section, we examine whether having unrestricted access to a bash terminal is really necessary to observe these gains, or whether simply having access to execution logs of the generated file is enough. For this experiment we compare *Bash* access with a simple method that executes and prints the output of the gistified file whenever it is edited (*Edit And Execute*). No other execution tools are available to the agent, including runtime information about the ground truth test. The results are surprising: having access to *fewer* tools actually increases performance. Indeed, we note that when give access to a full set of bash commands, the coding LLM tends to explore more tools, increasing the overall trajectory length, and potentially reaching the maximum step limit.

## 5.2 Tests with High Coverage are Harder to Gistify

In this section, we investigate what properties makes a given test hard to GISTIFY. We hypothesize that tests generating a longer and more complex execution trace would entail a harder task for the coding LLM. To this end, we investigate how two axes to measure a runtime execution's difficulty affect performance: the **length of trace**, as measured by the number of function calls executed, and the **number of unique files** touched by the tracing procedure. While these metrics correlate with one another, they will differ when, for example, a function is looped over many times or when the location of the relevant functions is in a single file versus across multiple files.

For this experiment, we use again the same configuration as prior analysis, namely Claude-4 with 50 tests sampled from the pylint codebase. In Figure 2a, we see a clear correlation between the difficulty of a given GISTIFY task, and how complex the execution traces are, according to both metrics considered. We leverage this insight to create a **GISTIFY-hard** subset, where we select the 30 most difficult examples according to each. We end up with 57 unique datapoints (30 from pylint, 28 from sklearn, 6 from seaborn). On this subset, performance drops to **21%**, as compared to **43%**, the baseline weighted performance average following the same distribution over repositories. Overall, this selection criteria offers a promising direction for designing challenging evaluation scenarios with GISTIFY.

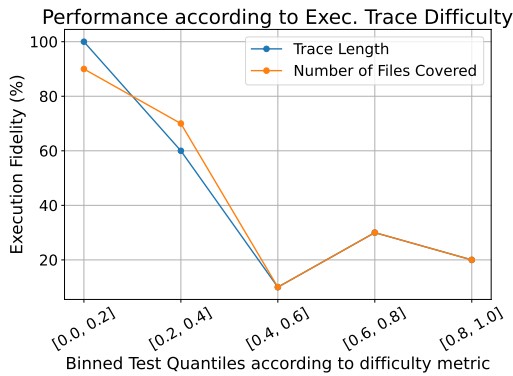
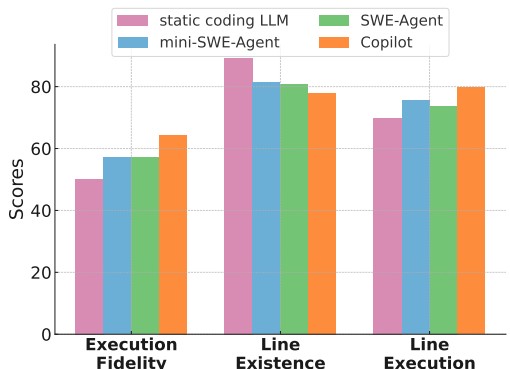

(a) Difficulty of the Gistify task is measured as a function of the execution trace difficulty of the underlying test.

(b) Performance of a static coding LLM and various agentic coding LLMs (mini-SWE-Agent, SWE-Agnet, Copilot).

## 5.3 STATIC CODING LLM

In this section, we experiment over how models perform in a static setup, where they have no access to tools and cannot iterate on the generated solution. As such static coding LLMs do not have tools, they cannot search or view files dynamically. Thereby, to measure a possible upper bound for non-agentic approaches, we provide as input all files that were accessed during the original program execution (gold files). Also, as they cannot iterate over multiple steps, they have to output everything at once and are therefore restricted by the context window of the LLM. Since solving the GISTIFY task involves touching multiple files, we observe in many cases that the inputs exceed the model's maximum sequence length. Thus, we sample a subset of test examples where the combined content fits within the 128K token limit of the LLM. As shown in Figure 2b, agentic models outperform static ones even when the latter receive all relevant files. This suggests that selecting files dynamically over multiple iterations is more effective than providing everything at once, which can overwhelm the model[5]. However, interestingly, the static coding LLM setup achieves the highest Line Existence score. This is likely because the model can copy lines directly from input, yet it performs worse on Line Execution and Execution Fidelity, suggesting that models do not have a good understanding of the codebase, often copying lines that are incomplete or incorrect. Additional details and chain-of-thought results are provided in Appendix E.2.

## 6 DISCUSSION AND CONCLUSION

In this paper, we introduced the GISTIFY task in which a coding LLM extracts a specific funtionality of a codebase into a single, self-contained file. Beyond serving as a standalone evaluation task that is easily applicable to arbitrary repositories with a test suite, the gistified file itself also opens several promising directions for research and practical applications. Large codebases often overwhelm automated agents due to their complex dependencies, and they especially struggle when tasked with fixing bugs that span multiple files (Ganhotra, 2025). In such scenarios, a gistified file would greatly reduce this challenge, and enable a more efficient reasoning about the codebase without navigating through unrelated code. In other words, this file could be leveraged in other downstream tasks such as code refactoring or debugging, or even as a way to extract and share a minimal implementation of a specific codebase functionality. Lastly, a current limitation of the results presented is the reliance on an existing test suite for a given repository. We believe that the GISTIFY task can be extended to arbitrary entrypoints, although issues stemming from non-deterministic execution will need to be carefully addressed. We defer a proper exploration of this to future work.

In summary, with coding LLMs increasingly being deployed in real-world software development, the need for automatically constructing evaluation setups that require codebase-level understanding of arbitrary repositories is growing. Through extensive experiments across a range of models and frameworks, we found that state-of-the-art LLMs still face challenges on the GISTIFY task, espe-

---

[5]See Appendix E.3 for detailed statistics on the usage of various tools.

cially when faced with long, complex execution traces. Our analysis shows that incorporating global code context or execution-aware tools improves performance, and agentic coding LLM tend to handle the task more effectively by reasoning about which files to inspect using various tools. Beyond serving as a benchmark, the gistified files themselves are valuable artifacts. They distill the essential functionality of complex systems into a compact, executable form, making them easier to inspect and understand. Such files could support a range of practical applications, including debugging, refactoring, and code review, which we leave this for future work.

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

# A    RELATED WORKS

## A.1    METHODS FOR CODEBASE-LEVEL UNDERSTANDING

Recent work on autonomous agents for codebase-level code understanding has focused on improving code navigation, reasoning, and generation through structured representations and planning. Approaches leverage structural information of code for function-call graphs, module-dependency graphs, and hierarchical code structures to provide models with core components of repositories (Wang et al., 2025; Liu et al., 2024). Another line of work integrate multi-step reasoning and state update policies to enable more effective planning over complex tasks (Bairi et al., 2024; Gautam et al., 2025). Additional methods combine various agents with multiple tools to streamline codebase-level exploration and task solving (Luo et al., 2024; Zhang et al., 2023; Shrivastava et al., 2023; Wang et al., 2024; Yang et al., 2024; Tang et al., 2023; aider, 2025; Microsoft, 2025; cursor, 2025).

# B    GISTIFY

## B.1    METRICS

**Execution Fidelity**    Execution fidelity measures whether the generated gistified file reproduces the same functional behavior as the original codebase under the given command. This includes producing the same number of test passes or failures, as well as consistent outputs and error handling. If the file's behavior matches the original codebase, it is assigned 100%; otherwise it receives 0%.

**Line Execution Rate**    The line execution rate measures the proportion of lines in the gistified file that are actually executed when running it under the given command. We first analyze the gistified file to identify which lines are executable (e.g., imports, function or class definitions) versus not-executable (e.g., comments). Using a tracing function, we then determine which of the executable lines are touched during execution. The line execution rate is computed as the fraction of executable lines that are executed. A rate of 100% indicates that the gistified file is concise and contains primarily necessary lines that are executed, while 0% indicates that non of the executable lines were touched. When calculating line execution rate, we exclude the tests where the self-containment is 0% as the goal of line execution rate is to evaluate the model's ability to construct concise, executable file, not to penalize failures in generating runnable code.

We classify each line of code into three categories: executable, potentially executable, and non-executable. Executable lines include imports and functional code that can be directly run. Potentially executable lines are those that may or may not be executed during a run, such as the except block of a try-except statement or placeholders for classes and function definitions. Non-executable lines, such as comments, are those that have no effect on execution. To calculate the line execution rate, we first classify each line in the gistified file and then consider only the *executable* lines. Non-executable lines are ignored since their presence or absence does not affect execution outcomes, and potentially executable lines are excluded because they are often ambiguous (e.g., placeholders) and cannot be reliably judged as necessary or removable.

**Line Existence Rate**    The line existence rate measures the proportion of lines in the gistified file that are directly preserved from the original codebase. We first parse both the gistified file and the original codebase into blocks, where each block corresponds to a class or function. Within classes, functions are nested under their parent class, forming a hierarchy. Lines outside of any block (e.g., top-level statements) are treated as standalone units.

For each block in the gistified file, we locate the corresponding block in the original codebase using its name and hierarchical position. If a matching block exists, we compare the two line by line to determine which lines are preserved; whether the lines in the gistified block appear in the corresponding original block. If no match is found, all lines in that block are treated as non-existent. For lines outside any block, existence is determined by direct comparison with top-level lines in the original codebase.

---

**Preprocess**

**Require:** Gistified file $G$, original repository $R$, module name $M$, execution information $E$ for lines in $G$

1: **Preprocessing: Line Classification**
2: **for all** lines $\ell$ in $G$ **do**
3:     assign $\text{type}[\ell] \in \{\texttt{comment}, \texttt{control\_flow}, \texttt{definition}, \texttt{executable}, \texttt{import}, \texttt{blank}\}$
4: **end for**

---

Figure 3: Details of Line Execution and Line Existence (Part 1): Preprocessing

---

**Details of Line Execution**

**Require:** Preprocessed gistified file $G$, original repository $R$, module name $M$, execution information $E$ for lines in $G$
**Ensure:** Line execution rate $r_{\text{exec}}$

1: **Line Execution Rate**
2: $S \leftarrow \{\ell \in G \mid \text{type}[\ell] \in \{\texttt{executable}, \texttt{import}\}\}$
3: $S_{\text{exec}} \leftarrow \{\ell \in S \mid \ell \text{ is marked as executed in } E\}$
4: $r_{\text{exec}} \leftarrow \frac{|S_{\text{exec}}|}{|S|}$

---

Figure 4: Details of Line Execution and Line Existence (Part 2): Line Execution

An existence rate of 100% indicates perfect preservation of the original code without hallucinated content.

**Normalization for Line-wise Code Matching** Figure 3, 4, and 5 show the detailed procedure to compute the line existence rate and line execution rate for a gistified file. The process begins by classifying every line of the gistified file into one of the categories: comment, control flow, definition, executable, import, or blank (Figure 3). This classification forms the basis for later filtering and comparison steps.

When calculating the *line execution rate*, we consider only lines classified as executable or import, because these are the lines whose execution status can be directly observed in the execution information (Figure 4). We deliberately exclude control-flow and definition lines from this rate: although they are crucial for the gistified file to behave like the original repository, whether such lines "execute" or not is often input- or parameter-dependent and therefore not reliably captured by a simple per-line execution count. The validity of control-flow and definition structures will be instead be indirectly assessed through our execution fidelity metric, which measures whether the gistified file and the original repository exhibit consistent overall execution behavior. In this way, the line execution rate focuses on directly measurable execution coverage, while execution fidelity provides a higher-level signal about behavioral correctness.

When calculating the *line existence rate*, both the gistified file and the original repository are parsed into nested structural blocks such as functions or classes. Any lines that are not contained within a structural block are treated separately as top-level lines (line 1-4 in Figure 5). For all cases, when calculating the line existence rate, we do not consider comments or blanks. The algorithm then examines each structural block in the gistified file and attempts to find the matching block from the original repository. If no matching block is found, all lines in the block are considered missing (line 12-19 in Figure 5). Otherwise (line 20-40 in Figure 5), each line undergoes a series of normalization steps such as fixing whitespace, indents, removing trailing comments, etc. The normalized line is then evaluated according to its type. Based on the comparison, each line in the block is marked as either existing or not. Once all lines have been processed, the line existence rate is computed by

---

**Details of Line Existence**

**Require:** Preprocessed gistified file $G$, original repository $R$, module name $M$, execution information $E$ for lines in $G$
**Ensure:** Line existence rate $r_{\text{exist}}$

1: **Structural Block Parsing**
2: parse $G$ and $R$ into nested structural blocks (e.g., functions and classes)
3: $\mathcal{B}_G \leftarrow$ set of blocks in $G$
4: $L_{\text{top}} \leftarrow$ lines in $G$ not contained in any structural block

5: **Initialization**
6: **for all** lines $\ell$ in $G$ **do**
7:     $\text{exists}[\ell] \leftarrow$ undefined
8: **end for**

9: **Block-Level Existence Analysis**
10: **for all** blocks $b \in \mathcal{B}_G$ **do**
11:     $\text{orig} \leftarrow$ best-matching block for $b$ in $R$
12:     **if** orig does not exist **then**
13:         **for all** lines $\ell \in b$ **do**
14:             **if** $\text{type}[\ell] \notin \{\texttt{comment},\texttt{blank}\}$ **then**
15:                 $\text{exists}[\ell] \leftarrow$ false
16:             **end if**
17:         **end for**
18:         **continue** to next block
19:     **end if**
20:     **for all** lines $\ell \in b$ **do**
21:         **if** $\text{type}[\ell] \in \{\texttt{comment},\texttt{blank}\}$ **then**
22:             **continue**
23:         **end if**
24:         normalize $\ell$: fix spacing, remove trailing comments, remove module prefixes, and split compound (";") statements
25:         **if** $\text{type}[\ell] = \texttt{control\_flow}$ **then**
26:             **if** $\ell$ is an $\texttt{if}$ or $\texttt{elif}$ statement **then**
27:                 set $\text{exists}[\ell]$ by comparing the conditional expression with corresponding $\texttt{if}/\texttt{elif}$ statements in orig
28:             **else if** $\ell$ is an $\texttt{else}$ statement **then**
29:                 $\text{exists}[\ell] \leftarrow$ true for else-body matches
30:             **end if**
31:         **else if** $\text{type}[\ell] = \texttt{definition}$ **then**
32:             verify existence of corresponding decorators (if any) in orig
33:             verify existence of each argument in the definition separately in orig
34:             assign $\text{exists}[\ell]$ based on these matches
35:         **else if** $\text{type}[\ell] = \texttt{import}$ **then**
36:             decompose into individual imports and set $\text{exists}[\ell]$ by per-import matching
37:         **else**
38:             compare against lines in orig and assign $\text{exists}[\ell]$ accordingly
39:         **end if**
40:     **end for**
41: **end for**

42: **Line Existence Rate**
43: $L_{\text{valid}} \leftarrow \{\ell \in G \mid \text{type}[\ell] \notin \{\texttt{comment},\texttt{blank}\}\}$
44: $L_{\text{exist}} \leftarrow \{\ell \in L_{\text{valid}} \mid \text{exists}[\ell] = \text{true}\}$
45: $r_{\text{exist}} \leftarrow \frac{|L_{\text{exist}}|}{|L_{\text{valid}}|}$

Figure 5: Details of Line Execution and Line Existence (Part 3): Line Existence

Table 4: Line execution performance of various models in the SWE-Agent framework with Gemini-2.5-pro as the LLM-as-Judge.

| Model | Original Evaluation | LLM-as-Judge |
|---|---|---|
| GPT-5-mini | 74.8 | 73.5 |
| GPT-5 | 81.7 | 81.3 |
| Claude-3.7 | 69.9 | 69.1 |
| Claude-4 | 72.9 | 71.8 |

how many lines are marked as existing in the original repository over all lines except for comments or blanks (line 43-45 in Figure 5).

**Line Execution Performance when Adding LLM-based judge module** Table 4 shows line execution performance when incorporating LLM-as-Judge module. Adding such a module allows us

Table 5: Details of the GitHub repositories used as the test set.

| Repository | URL | License |
|---|---|---|
| flask | `https://github.com/pallets/flask` | BSD 3-Clause |
| requests | `https://github.com/psf/requests` | Apache-2.0 |
| pylint | `https://github.com/pylint-dev/pylint` | GPL 2.0 |
| scikit-learn | `https://github.com/scikit-learn/scikit-learn` | BSD 3-Clause |
| seaborn | `https://github.com/mwaskom/seaborn` | BSD 3-Clause |
| debug-gym | `https://github.com/microsoft/debug-gym` | MIT |

to evaluate not only *executable and import lines* but also *control-flow and definition lines* (e.g., try:, else:), which were skipped in the standard metric because they exist as structural lines rather than lines actually executed (Details in previous paragraph).

Given the tracing output for each line, the LLM determined whether these lines would be executed according to Python semantics and runtime control flow. A line is marked as *executed* if the interpreter enters it along any execution path, and *not executed* if it is unreachable. Following Jiang et al. (2025), we use Gemini-2.5-pro as the Judge model, as it shows strong performance on code-related reasoning tasks.

We re-evaluated the results using the SWE-Bench framework with the LLM-as-Judge module added. We observed that including these additional lines slightly decreases the line execution rate, which we attribute to rarely executed lines, such as except:, being taken into account. The largest drop observed was 1.3% (GPT-5-mini). Overall, however, the relative trends across models remain consistent. Further analysis shows that control-flow and definition lines make up approximately 4.1% of lines in a gistified file, suggesting why adding the LLM-based judge has minimal impact on the metric.

**User Study on the Definition** We conduct a user study to assess whether our proposed metrics, line existence and line execution, successfully capture the task's intended notions of *faithfulness* and *minimality*, respectively.

We recruit three software/AI engineers as annotators and provide them with 15 test cases. Each test case contains a pair of gistified files generated by different frameworks or different models. For each pair, annotators were asked to choose the file that better satisfied the task's criteria of minimality and faithfulness. Annotators were given: (1) a description of how the gistified files were constructed, (2) the definitions of minimality and faithfulness used in our task, (3) the two gistified files for comparison, and (4) execution-tracing information to help them understand the flow of each test run. Figure 6 shows the instructions provided to annotators.

To measure alignment between human judgment and our metrics, we computed Cohen's kappa correlation between the annotators' selections and the rankings produced by line existence and line execution. We observed an average Cohen's kappa of 0.61 (0.52, 0.72, 0.58) for minimality with line execution and 0.76 (0.81, 0.71, 0.77) for faithfulness with line existence, indicating that our metrics correspond well with human judgments of minimality and faithfulness.

## C  EXPERIMENTAL SETTING

### C.1  FRAMEWORK

We evaluate experiments with three agentic frameworks: mini-SWE-Agent (Yang et al., 2024), SWE-Agent (Yang et al., 2024), and Copilot (Microsoft, 2025). Unless otherwise noted, all experiments are run in the default GISTIFY setup, where the model is restricted from executing any commands (e.g., `python`, `pytest`). SWE-Agent and Copilot Agent enable LLMs to interact with a codebase through a suite of tools, including bash commands. These tools support capabilities such as viewing, searching, editing, and creating files or directories. In addition, Copilot Agent extends this functionality with browser integration, explicit reasoning, and API usage. mini-SWE-agent is a simplified variant of SWE-Agent that only supports bash commands. Despite its minimal design, it achieves strong performance on the SWE-Bench Verified benchmark (Jimenez et al., 2023). For

**User Study Instruction**

You are given two code files produced by a coding LLM. Both files attempt to complete the same task: create a single, minimal, self-contained file that reproduces a specific functionality of a codebase.

This is the prompt we provide to the model to describe the task:

[Figure C.3]

Your task:
Evaluate the two files and determine which one is more minimal and more faithful according to the criteria below.

1. A file is "minimal" if:
- It contains only the code truly required to reproduce the runtime behavior.
- Unused functions, classes, variables, or imports should be removed.
- The evaluator must understand the code well enough to identify which lines are actually executed and are essential. However, we do not penalize minor extra lines that exist solely because of formatting or structure constraints (e.g., keeping an unused method such as try-except because the format requires it).

2. A file is "faithful" if:
- No hallucinated code is introduced. Every piece of code must come directly from the original codebase.
- The structure of the code must stay consistent with the original. For example, moving a method that originally lived inside a class to the top level is considered incorrect.
- Simple changes such as incorrect indentation or broken multi-line statements are not penalized as long as the intended semantics are still clear. However, structural changes that alter the meaning of the code are penalized. For example: If the original code defines a class with inheritance, such as class ABC(DE):, but the generated file changes it to class ABC:, then this is considered incorrect, because removing the parent class changes the actual operation of the code.

Figure 6: Instruction for User Study

both mini-SWE-Agent and SWE-Agent, we set the maximum number of steps to 50 and run them in the same Docker environment, using the current version of the repositories.

## C.2 EXPERIMENTAL TEST SET CONSTRUCTION

Table 5 summarizes the repositories used in our evaluation. For each repository, we begin by extracting all available test cases, including parameterized ones. For experimental test runs, we group tests[6] that share the same base structure but differ only in parameterization, treating them as a single test. During evaluation, however, we execute all parameterized instances and measure how many are passed, thereby assessing execution fidelity. Finally, we filter out environment-dependent tests, such as those requiring relative file paths or fixed module locations. In the main experiments, we used 25 test instances for each of the six codebases, and the analysis was conducted using 50 test instances from the pylint codebase.

## C.3 PROMPT FOR GISTIFY

Figure 7 shows the prompt used in the main experiments.

## C.4 PROVIDING SPECIFIC PARAMETERS TO COMMANDS TENDS TO MAKE MODELS GENERATE PARAMETER-SPECIFIC GISTIFIED FILES

We observe that when specific command-line parameters are provided, models often adapt the generated gistified file to those parameters rather than producing a fully general solution. Examples of this parameter-specific behavior are shown in Figures 8 and 9. Accordingly, in our experiments, we group test cases based on the parameters provided to the command.

## C.5 ADDITIONAL HYPERPARAMETERS

All experiments were executed with the same base set of hyperparameters. For framework specific hyperparameters (e.g. cache management, system prompts, tools), we use the default configuration made available by each framework, except for vscode-copilot-chat, where we have listed the tools used in the experiments can be found in 6.

All the models used in the were configured with a maximum context window length of 128K. We used a default temperature of 1 and performed a single run for each configuration.

# D RESULTS

## D.1 EXAMPLE OF GISTIFIED FILE

Figure 10 and Figure 11 show two gistified files on the same test case with different models; each model succeed or fail in generating a gistified file with execution fidelity of 100% and 0%, respectively. In the successful case (Figure 10), the generated file handles both parameters correctly, achieving a 100% line existence rate, a 65.5% execution rate, and a test $F_1$ score of 100. In contrast, the failed case (Figure 11) cannot execute due to a missing `import pytest` statement. Moreover, the hallucinated test function yields a test $F_1$ score of 0, and the file shows a much lower line existence rate of 28%.

## D.2 ERROR ANALYSIS OVER EXECUTION FAILURE

We categorize errors into four types:

**Import Error** Figure 12 shows an example of *Import Error*. This occurs when the model incorrectly imports the original repository (e.g., `import requests`) instead of inlining the required modules into the gistified file.

---

[6]We adopt this grouping design as we observe that models often overfit to specific values when parameters are provided. See Appendix C.4 for more details.

---

**Prompt for GISTIFY**

I've uploaded a python code repository in the directory {working dir}.
There is an original test invocation (the command that reproduces behavior we want to preserve): problem statement Your job: create a single file named 'concise.py' saved at {working dir} that is **self-contained**, **minimal**, and **executable**, and when used in place of the original test run reproduces the same runtime behavior and outputs. Follow the instructions below when creating the file.

OUTPUT
- Produce one file only: '{working dir}/concise.py'.
- The assistant must return only the contents of 'concise.py' (no extra files, no analysis, no commentary).

HIGH-LEVEL RULES for creating 'concise.py'
1. Inline internal dependencies
* Copy into 'concise.py' every function, class, or top-level code from the files inside {working dir} that is executed when running {problem statement}.
* Do not use 'import' statements for modules defined in {working dir}.
2. Remove unexecuted lines
* When copying lines in 'concise.py', keep only the lines that is actually executed when running {problem statement}.
* Delete unused functions, classes, variables, if-else, imports, and unreachable branches.
* Ensure the file remains syntactically correct and minimal after removal.
3. Preserve original source lines
* Do not rewrite or reformat lines unless necessary to keep the files valid.
* Do not arbitrary generate new lines that do not exist in the original {working dir} files.
* You may adjust indentation, remove empty 'else'' blocks, or adapt 'try-except' structures only when required to preserve correctness.
4. Keep external imports
* Leave imports to external libraries, frameworks, or standard runtime libraries unchanged.
* Only remove or inline dependencies that come from {working dir}.
5. No shortcuts or cheating
* Do not stub, fake, or monkey-patch external modules.
* Do not reimplement or newly add third-party libraries.
* Do not hard-code outputs
* Do not replace test logic with simplified equivalents
6. Preserve test behavior
* The test function much remain unchanged, except for import adjustments needed to reference inlined code.
* The output, exceptions, or exit codes must match the original run of {problem statement}.
7. Do not execute the code
* Do not run or simulate the program (e.g., with 'pytest', 'python', or any other tools)

Figure 7: Base Prompt Template for GISTIFY Task.

**File Creation Failure**  This error arises when the model fails to generate the gistified file. This can happen in two ways: (1) the model exceeds the maximum step limit or (2) the model completes within the time limit but still fails to generate the new file using the tool.

**Missing Test Function**  This occurs when the generated gistified file does not contain the modules for specified test in the given command. It typically arises when the model fails to locate or copy the modules necessary for the test into the gistified file. Conceptually, this corresponds to a 0% line existence rate for the test function. Since the presence of the modules for the given test case is essential for validation, we classify this as an error.

```python
@pytest.mark.parametrize(
    "value, expected",
    (
        ("application/xml", ("application/xml", {})),
        (
            "application/json ; charset=utf-8",
            ("application/json", {"charset": "utf-8"}),
        ),
        ("text/plain", ("text/plain", {})),
        ...
)
def test__parse_content_type_header(value, expected):
    assert _parse_content_type_header(value) == expected
```

(a) Original Test Case

```python
def test__parse_content_type_header():
    """Test for the _parse_content_type_header function with application/
    json and charset=utf-8"""
    value = "application/json ; charset=utf-8"
    expected = ("application/json", {"charset": "utf-8"})
    assert _parse_content_type_header(value) == expected
```

(b) Gistified File

Figure 8: Example of a model generating a parameter-specific gistified file when given a command that includes a parameter.

We also observe an interesting behavior of GPT-5 where it tends to insert `__name__ == "__main__"` even though it is not provided in the original codebase and even though it is explicitly mentioned that we will test on the provided command and expect the same output. They often remove the test function but move the lines in the test function under the `"__main__"` guard (e.g., Figure 21). We hypothesize that this may be because they are more familiar with codebases following this pattern. We also observe cases where the model attempts to "cheat" the task by injecting a mock, in-memory version of the original codebase package to satisfy import dependencies, rather than copying the necessary code inline (e.g., Figure 23).

**Pytest Runtime Error**

This error refers to failures that occur during pytest execution, such as syntax errors or fixture-related issues (e.g., Figure 13). Although the absence of test functions is also one of pytest failures, we explicitly separate those cases by first verifying the presence of the required test functions and running pytest only when they exist.

### D.3 Tools Available in GitHub Copilot

Table 6 shows the list of available tools in Github Copilot.

### D.4 Change Test

even high performing models and frameworks (especially GPT-5 and GPT-5-mini) seems to modify test codes even though explicitly mentioned not to. We observed three common modification: (1) removing the test function but move the lines in the test function under the `"__main__"` guard (e.g., Figure 21), (2) adding the `"__main__"` guard even though unnecessary (e.g., Figure 22), and (3) mocking a minimal in-memory package to bypass missing dependencies and force the test to run (e.g., Figure 23).

```
@pytest.mark.parametrize(
    "url, expected",
    (
        ("http://192.168.0.1:5000/", True),
        ...
        ("http://google.com:5000/v1.0/", False),
    ),
)
def test_should_bypass_proxies_no_proxy(url, expected, monkeypatch):
    """Tests for function should_bypass_proxies to check if proxy
    can be bypassed or not using the 'no_proxy' argument
    """
    no_proxy = "192.168.0.0/24,127.0.0.1,localhost.localdomain,172.16.1.1
    "
    # Test 'no_proxy' argument
    assert should_bypass_proxies(url, no_proxy=no_proxy) == expected
```

(a) Original Test Case

```
def test_should_bypass_proxies_no_proxy(url, expected, monkeypatch):
    """Tests for function should_bypass_proxies to check if proxy
    can be bypassed or not using the 'no_proxy' argument
    """
    no_proxy = "192.168.0.0/24,127.0.0.1,localhost.localdomain,172.16.1.1
    "
    # Test 'no_proxy' argument
    assert should_bypass_proxies(url, no_proxy=no_proxy) == expected
```

(b) Gistified File

Figure 9: Example of a model generating a parameter-specific gistified file when given a command that includes a parameter.

### D.5 ADDITIONAL METRICS

Table 7 shows the result of additional evaluation metrics, including the *Average Pytest Pass Rate*, which is defined as the average test pass rate over cases with at least one successful run, and the *Test $F_1$ Score*, which quantifies the line-wise $F_1$ existence between the test functions in the original codebase and those in the gistified fie.

GPT-5 shows a notably higher Average Pytest Pass Rate, indicating that among the ones they successfully generate, they tend to pass all pytest. For the Test $F_1$ Score, Claude-4 shows the highest performance, aliging with the trend discussed in Section 4.4.

### D.6 PERFORMANCE ON LARGER SAMPLE SIZE

Table 8 shows the result of GPT-5-mini with SWE-Agent framework when running over larger number of samples (16.7x more samples from the 6 repositories). We observe that these extended results align with our initial numbers, showing that our initial sample size provides a meaningful estimate of the performance for a given configuration.

## E ANALYSIS

### E.1 EFFECT OF VARIOUS STRATEGIES AND TOOLS

**Prompt-Based Guidance** We experiment with two variants of the prompt, *Reading* and *Tracing*, where, on top of the base prompt (Figure 7), we add specific instructions of `How to Operate` to encourage reasoning using a particular strategy. The addition prompt detail of *Reading* is in Figure 15, and for *Tracing* is in Figure 16.

```
# Licensed under the GPL: https://www.gnu.org/licenses/old-licenses/gpl-2.0.html
# For details: https://github.com/pylint-dev/pylint/blob/main/LICENSE
# Copyright (c) https://github.com/pylint-dev/pylint/blob/main/CONTRIBUTORS.txt

from __future__ import annotations

import os
from collections.abc import Sequence
from typing import Any

import pytest

def discover_package_path(modulepath: str, source_roots: Sequence[str]) -> str:
    """Discover package path from one its modules and source roots."""
    dirname = os.path.realpath(os.path.expanduser(modulepath))
    if not os.path.isdir(dirname):
        dirname = os.path.dirname(dirname)

    # Look for a source root that contains the module directory
    for source_root in source_roots:
        source_root = os.path.realpath(os.path.expanduser(source_root))
        if os.path.commonpath([source_root, dirname]) in [dirname, source_root]:
            return source_root

    # Fall back to legacy discovery by looking for __init__.py upwards as
    # it's the only way given that source root was not found or was not provided
    while True:
        if not os.path.exists(os.path.join(dirname, "__init__.py")):
            return dirname
        old_dirname = dirname
        dirname = os.path.dirname(dirname)
        if old_dirname == dirname:
            return os.getcwd()

@pytest.mark.parametrize(
    "py_mod_base_name",
    ("__init__", "impl"),
    ids=("explicit-namespace", "implicit-namespace"),
)
def test_discover_package_path_source_root_as_parent(
    py_mod_base_name: str,
    tmp_path: Any,
) -> None:
    """Test discover_package_path when source root is a parent of the module."""
    # Create this temporary structure:
    # /tmp_path/
    #    project/
    #        my-package/
    #            __init__.py
    project_dir = tmp_path / "project"
    package_dir = project_dir / "mypackage"
    package_dir.mkdir(parents=True)
    (package_dir / f"{py_mod_base_name}.py").touch()

    # Test with project_dir as source root (parent of package)
    result = discover_package_path(str(package_dir), [str(project_dir)])
    assert result == str(project_dir)
```

Figure 10: Example of a successful gistified file: the code correctly handles all parameters, achieving 100% line–existence rate, a test $F_1$ score of 100, and an execution rate of 65.5%.

**Global Information via Tools** We experiment with two tools that provide global information: *RepoGraph* and *Tracing*. Details of the information provided to the model about each tool are shown in Figure 17.

*RepoGraph* (Ouyang et al., 2024) is a plug-in module designed to help LLMs leverage the codebase-level structure. It parses code at the line level, extracts relationships, and constructs a graph where each node represents a line of code and each edge encodes dependencies between code definitions and their references. Thereby, when given a specific module, it returns the relationship with other modules as represented within the constructed graph.

```python
import os
import sys
from contextlib import import contextmanager
from pathlib import Path

def discover_package_path(path, _search_paths):
    """
    Minimal version: accept a filesystem path (string or Path) and return a Path
    to that directory. This matches the usage in the specific test where a
    path to the tests data directory is passed.
    """
    if isinstance(path, Path):
        p = path
    else:
        p = Path(path)
    if p.exists() and p.is_dir():
        return p
    raise ValueError("Cannot determine package path")

@contextmanager
def augmented_sys_path(paths):
    """
    Minimal context manager: set sys.path to the parent directories of the
    provided paths (in order), restoring the original sys.path on exit.
    """
    original = list(sys.path)
    try:
        new = []
        for p in paths:
            pth = Path(p)
            # Ensure we use the parent directory (the "source root" in the test)
            parent = pth.parent
            new.append(str(parent))
        sys.path[:] = new
        yield
    finally:
        sys.path[:] = original

@pytest.mark.parametrize(
    "py_mod_base_name",
    ("__init__"),
    ids=("explicit-namespace"),
)
def test_discover_package_path_source_root_as_parent():
    # This test's behavior is preserved: it uses discover_package_path on the
    # tests data directory and ensures augmented_sys_path sets sys.path to its
    # parent (the project/tests directory).
    TEST_DATA_DIR = os.path.abspath(os.path.join(os.path.dirname(__file__), "tests", "data"))
    PROJECT_ROOT_DIR = os.path.abspath(os.path.join(TEST_DATA_DIR, ".."))
    with augmented_sys_path([discover_package_path(TEST_DATA_DIR, [])]):
        assert sys.path == [PROJECT_ROOT_DIR]
```

Figure 11: Example of failed gistified file: the code fails to import `pytest`. The model hallucinates the function `test_discover_package_path_source_root_as_parent()`, resulting in a test $F_1$ score of 0 and a low line–existence rate of 28.0%

*Tracing* is a tool that uses the tracer provided from the `sys` module to execute a command and track which components of the codebase are accessed. When the model uses the tool with a specific command, the tool provides the model with the files and functions touched when running the command, in the order in which they are encountered.

**Execution-Based Tools**  We experiment with two execution-based tools: the *Bash* tool and the *Edit and Execute* tool.

The *Bash* tool is a basic utility that allows the model to invoke any necessary Bash commands. In contrast, the *Edit and Execute* tool is designed specifically for working with the gistified file: it enables the model to create or modify the gistified file and optionally execute it to verify changes.

The primary difference between the two tools is their scope of execution. The *Bash* tool can run commands on both the original codebase and the gistified file, whereas the *Edit and Execute* tool is restricted to executing only the gistified file.

```
@click.option("--all-methods", is_flag=True, help="Show HEAD and OPTIONS
    methods.")
@with_appcontext
def routes_command(sort, all_methods):
    """Show all registered routes with endpoints and methods."""
    from flask import current_app

    rules = list(current_app.url_map.iter_rules())

    if not rules:
        click.echo("No routes were registered.")
        return
```

Figure 12: Example of an *Import Error*: the gistified file imports from the original repository (e.g., `from flask import current_app`).

```
T = t.TypeVar("T")

class ConfigAttribute(t.Generic[T]):
    """Makes an attribute forward to the config"""

    def __init__(
        self, name: str, get_converter: t.Callable[[t.Any], T] | None =
    None
    ) -> None:
        self.__name__ = name
        self.get_converter = get_converter
```

(a) Original Test Case

```
class ConfigAttribute:
    def __init__(
        self, name: str, get_converter: t.Callable[[t.Any], T] | None =
    None
    ) -> None:
        self.__name__ = name
        self.get_converter = get_converter
```

(b) Gistified File

Figure 13: Example of an *Pytest Runtime Error*: gistified file fails with error message E TypeError: type 'ConfigAttribute' is not subscriptable

We include an example of the behavior observed when adding the execution tool in Figure 18. Common patterns we observe are: (1) the model first runs the provided command to identify which files are accessed and to gather execution feedback; (2) after creating a file, it iteratively executes it to verify that the generated gistified file behaves as expected; and (3) it repeatedly compares the outputs of the gistified file and the original codebase under the given command. We also observe that, due to this iterative checking process, enabling the execution tool often leads the model to terminate because it reaches the maximum step limit.

## E.2  STATIC CODING LLM

Figure 19 presents the performance of static and dynamic coding LLM, including a static model augmented with a *single-turn, multi-step chain-of-thought* prompt. To test whether explicit reasoning and planning can mitigate the limitations of static coding LLMs, which must output the entire program in a single turn, we compare this CoT-augmented static model against both the baseline static model and the dynamic models.

| Tool | Description |
|------|-------------|
| copilot_getNotebookSummary | Returns the list of Notebook cells with id, types, line ranges, language, execution info, and output mime types. Useful for getting cell IDs, execution order, and outputs. |
| edit_notebook_file | Edit an existing Notebook file in the workspace. Supports inserting, deleting, or editing cells while preserving whitespace and indentation. |
| apply_patch | Edit text files using a special diff/patch format. Do not use for Jupyter notebooks. |
| semantic_search | Run a natural language search for relevant code or documentation comments in the workspace. |
| create_directory | Create a new directory structure in the workspace (like `mkdir -p`). |
| create_file | Create a new file with specified content. Automatically creates directories if they do not exist. |
| file_search | Search for files in the workspace by glob pattern (e.g., `**/*.js`). Returns matching paths only. |
| test_search | For a source file, find the corresponding test file, and vice versa. |
| grep_search | Fast text or regex search in the workspace. Useful for exact string or regex queries. |
| run_notebook_cell | Run a code cell in a notebook file and return the output. Avoid running Markdown cells. |
| read_notebook_cell_output | Retrieve the latest output for a notebook cell, even if not run in the current session. |
| get_search_view_results | Returns results from the search view. |
| github_repo | Search a GitHub repository for relevant code snippets. Use only for external repos, not local workspaces. |
| insert_edit_into_file | Insert or edit code in an existing file using minimal hints, avoiding duplication of unchanged code. |
| install_extension | Install an extension in VS Code. Used only during workspace creation. |
| list_dir | List the contents of a directory (folders and files). |
| create_new_jupyter_notebook | Generate a new Jupyter Notebook (.ipynb) in VS Code. |
| create_new_workspace | Set up a complete new project (scaffolding, dependencies, config, boilerplate). |
| get_project_setup_info | Provides project setup information for a VS Code workspace after workspace creation. |
| read_file | Read the contents of a file. Supports offsets and limits for large files. |
| open_simple_browser | Preview or open a URL in VS Code's Simple Browser. |
| test_failure | Include test failure information in the prompt. |
| think | Think deeply about a request and log structured reasoning (no execution). Useful for planning, debugging, and brainstorming. |
| get_vscode_api | Retrieve comprehensive VS Code API documentation and references for extension development. |
| run_vscode_command | Run a VS Code command by ID with arguments. Used mainly in workspace creation. |
| fetch_webpage | Fetch main content from a webpage for summarization or analysis. |

Table 6: Available tools and their descriptions. We note that many tools available to the agent are never used.

Table 7: Average Pytest Pass Rate and Test $F_1$ Score of different models using SWE-Agent on the main table (Table 1) test dataset.

| Models | Execution Fidelity | Average Pytest Pass Rate | Test $F_1$ Score |
|--------|--------------------|--------------------------|-------------------|
| GPT-5-mini | 30.9 | 49.2 | 47.9 |
| GPT-5 | 30.7 | 88.8 | 45.0 |
| Claude-3.7 | 40.7 | 61.9 | 55.9 |
| Claude-4 | 56.7 | 72.2 | 60.0 |

Table 8: Performance of GPT-5-mini with SWE-Agent framework on larger number of samples (2.5k).

| # of Test Instances | Execution Fidelity (w/ exec) | Line Existence | Line Execution |
|---------------------|------------------------------|----------------|----------------|
| 150 | 45.3 | 47.9 | 74.8 |
| 2.5k | 46.4 | 47.8 | 75.6 |

Overall, **we observe a trend similar to the baseline static model without a chain-of-thought**: adding chain-of-thought yields only a modest improvement over the static model without it, and it

```
@pytest.mark.parametrize(
    "value, expected",
    (
        ('foo="is a fish", bar="as well"', {"foo": "is a fish", "bar": "
    as well"}),
        ("key_without_value", {"key_without_value": None}),
    ),
)
def test_parse_dict_header(value, expected):
    assert parse_dict_header(value) == expected
```

(a) Original Test Case

```
assert parse_dict_header('foo="is a fish", bar="as well"') == {"foo": "is
    a fish", "bar": "as well"}
assert parse_dict_header("key_without_value") == {"key_without_value":
    None}
```

(b) Gistified File

Figure 14: The test function `test_parse_dict_header` is simplified: in the original, it used `@pytest.mark.parametrize` to feed multiple input/expected pairs into one function; in the gistified version, this is replaced with two direct `assert` statements, one per case.

---

**Behavior Reading**

How to Operate:
1. Examine the test file and the test function used for {problem statement}
2. Identify which module used by these functions are defined in {working dir}
3. Copy and inline the code from those modules into 'concise.py'
4. Check these modules for any internal functions or classes and inline them as needed.
5. Repeat this process recursively until all internal dependencies are inlined.
6. Do not forget to copy and paste external imports.

Figure 15: Prompt for *Reading* strategy.

---

**Trace Reasoning**

How to Operate:
1. Predict the execution traces.
2. Follow the traces and inline (copy) only the necessary executed lines into 'concise.py'
3. Repeat until all traces are fully handled.

Figure 16: Prompt for *Tracing* strategy.

---

**Tool Information**

* search repo (repograph): searches in the mentioned repository with a specific function or class, and returns the def and ref relations for the search term.
* trace info (tracing): returns the trace log when running the 'pytest' command for the test currently being debugged in the given repo

Figure 17: Details of the information provided to the model about each tool.

```
 {
   "role": "assistant",
   "tool_calls": {
     "arguments": "{\"command\":\"cd /app && python -m pytest concise.
py::TestDeprecationOptions::test_new_names -v\"}",
     "name": "bash"
     }
 },
 {
   "role": "tool",
   "content": [
     {
       "type": "text",
       "text": "OBSERVATION: test session starts ...AssertionError\n
...pylint: error: unrecognized arguments: /app/regrtest_data/empty.py
\n FAILED concise.py::TestDeprecationOptions::test_new_names ..."
     }
 },
 ...
 (fix concise.py based on observation)
 ...
 {
   "role": "assistant",
   "tool_calls": {
     "arguments": "{\"command\":\"cd /app && python -m pytest tests/
config/test_argparse_config.py::TestDeprecationOptions::
test_new_names -v -s\"}",
     "name": "bash"
     }
 },
 ...
 {
   "role": "assistant",
   "content": "Excellent! The messages are identical. Let me verify my
 implementation removes all unused imports and only includes what's
needed:"
 }
```

Figure 18: Trajectory when including execution tool. Models often *iteratively* execute the given codebase or generated gistified file to ensure that it operates expectedly.

Table 9: Analysis of tool usage during the GISTIFY task

| Models | Avg. tool usage | view | search | execute | other |
|---|---|---|---|---|---|
| GPT-5-mini | 10.8 | 71.9 | 9.8 | 1.7 | 16.6 |
| GPT-5 | 18.5 | 72.4 | 8.3 | 3.3 | 16.1 |
| Claude-Sonnet-3.7 | 17.3 | 67.5 | **10.1** | 4.5 | **17.9** |
| Claude-Sonnet-4 | **19.3** | **74.6** | 2.1 | **11.8** | 11.5 |

still performs worse than the dynamic models, underscoring the importance of multi-turn generation and tool use.

Relative to the baseline static model, the CoT-augmented version slightly improves line-execution rates but slightly decreases line-existence. We hypothesize that this is because the CoT procedure encourages the model to focus on minimal set of necessary lines.

Details of the prompt used in this experiment are provided in Figure 20.

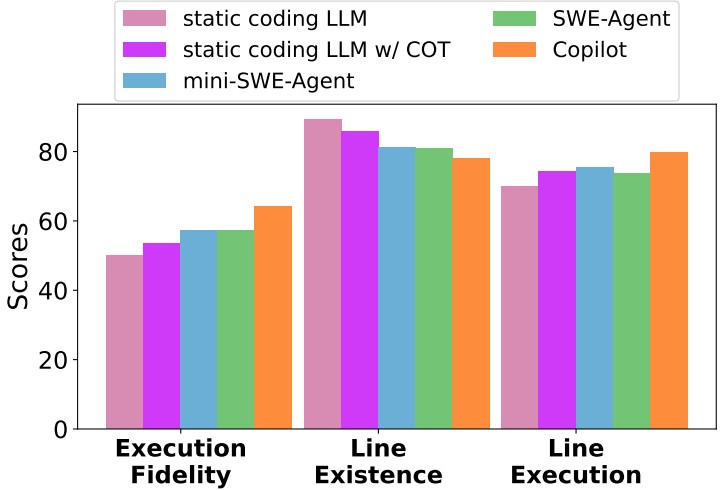

Figure 19: Performance over Static and Dynamic Coding LLM

---

**Static Coding LLM with CoT**

THINK STEP BY STEP, and SHOW YOUR REASONING. You must follow a multi-step solution process: (1) determine what code executes during test, (2) determine the minimal set of lines required, (3) plan the final single-file layout, and finally (4) output the new single-file code.

---

Figure 20: Prompt added to the static coding LLM to enable chain-of-thought reasoning. A similar instruction is also included in the system prompt.

```python
class TestGetNetrcAuth:
    def test_works(self, tmp_path, monkeypatch):
        netrc_path = tmp_path / ".netrc"
        monkeypatch.setenv("NETRC", str(netrc_path))
        with open(netrc_path, "w") as f:
            f.write("machine example.com login aaaa password bbbb\n")
        auth = get_netrc_auth("http://example.com/thing")
        assert auth == ("aaaa", "bbbb")
```

(a) Original Test Case

```python
if __name__ == "__main__":
    # Reproduce tests/test_utils.py::TestGetNetrcAuth::test_works
    with tempfile.TemporaryDirectory() as tmpdir:
        netrc_path = os.path.join(tmpdir, ".netrc")
        os.environ["NETRC"] = netrc_path
        with open(netrc_path, "w") as f:
            f.write("machine example.com login aaaa password bbbb\n")
        auth = get_netrc_auth("http://example.com/thing")
        assert auth == ("aaaa", "bbbb")
```

(b) Gistified File

Figure 21: Test Modification Case 1: The test `TestGetNetrcAuth.test_works` is converted from a pytest unit test into a standalone script.

```
# Test class and method - preserved unchanged
class TestArgparseOptionsProviderMixin:
    """Tests for the argparse implementation of OptionsProviderMixIn.

    The logger checker is used as an example checker for this
    implementation.
    """

    @staticmethod
    def test_logger_without_options() -> None:
        """Check that we raise messages when we do not supply any options
    ."""
        with pytest.raises(SystemExit) as ex:
            Run([LOGGING_TEST])
        assert ex.value.code == 2

# Main execution for pytest
if __name__ == "__main__":
    test = TestArgparseOptionsProviderMixin()
    test.test_logger_without_options()
```

Figure 22: Test Modification Case 2: Adding unnecessary "__main__" guard

```
# Create a minimal in-memory 'requests' package with required submodules.
requests_mod = types.ModuleType('requests')
requests_mod.__path__ = []
compat_mod = types.ModuleType('requests.compat')
structures_mod = types.ModuleType('requests.structures')

# Populate compat with only what's needed by this test suite import paths
    .
compat_mod.Mapping = Mapping
compat_mod.MutableMapping = MutableMapping
compat_mod.urljoin = urljoin

# Populate structures with the classes.
structures_mod.CaseInsensitiveDict = CaseInsensitiveDict
structures_mod.LookupDict = LookupDict

# Wire the package hierarchy and register in sys.modules.
requests_mod.compat = compat_mod
requests_mod.structures = structures_mod
sys.modules['requests'] = requests_mod
sys.modules['requests.compat'] = compat_mod
sys.modules['requests.structures'] = structures_mod

if __name__ == '__main__':
    import pytest
    raise SystemExit(pytest.main(['-q', 'tests/test_structures.py::
    TestCaseInsensitiveDict::test_list']))
```

Figure 23: Test Modification Case 3: Manually mocking a minimal in-memory package to bypass missing dependencies and force the test to run.

### E.3 TOOL USAGE RATES

Table 9 shows the statistics on tool usage across models using SWE-bench. We group various tools into four categories: view, search, execute, and other, which includes all remaining tools. For all

models, we compute usage rates both with and without execution enabled, and then average across the two settings.

Among all models, Claude-4 exhibits the highest average tool usage for each test cases, followed by GPT-5, Claude-3.7, and GPT-5-mini. In terms of specific functionality, Claude-4 shows the highest rate of both view and execute tool usage, while Claude-3.7 shows the highest usage of the search tool. To generate a high-quality gistified file, a model must effectively view relevant files and copy only the necessary content. The strong performance of Claude-4 on line existence may be related to its high usage of the *view* tool. Also, the *execution* tool tends to support correctness verification of the generated file, which would lead to high execution fidelity.

