# OpenReview forum: "Gistify: Codebase-Level Understanding via Runtime Execution"
_ICLR.cc/2026/Conference — ICLR 2026 Poster_

### Official Review · Reviewer_WQmg · 2025-10-27

**Soundness:** 2
**Presentation:** 3
**Contribution:** 2
**Rating:** 4
**Confidence:** 4

**Summary:**

This paper introduces GISTIFY, a novel task designed to evaluate code generation models on codebase-level understanding through runtime execution. Given a codebase and an entry-point command, the goal is to generate a single, self-contained, and minimal gistified file that reproduces the original runtime behavior. The authors formalize three evaluation metrics, Execution Fidelity, Line Execution Rate, and Line Existence Rate, to measure model performance. Experiments are conducted across multiple frameworks and state-of-the-art models on popular Python repositories. Results show that even advanced models struggle with complex execution traces, while agentic frameworks and access to execution-aware tools yield consistent improvements. The paper argues that GISTIFY offers a lightweight, reproducible benchmark that reflects real-world developer workflows and provides valuable distilled code artifacts for downstream applications.

**Strengths:**

- The paper formalizes a runtime-based approach to codebase understanding, which is underexplored in prior benchmarks.
- The three metrics (Execution Fidelity Rate, Line Execution Rate, Line Existence Rate) together provide a comprehensive evaluation of the GISTIFY task, offering a well-rounded assessment of the model’s effectiveness.
- Experiments cover multiple models and frameworks, providing comparative insights.

**Weaknesses:**

- The Execution Fidelity can only be signed to 1 or 0, which lacks granularity in capturing partially correct implementations.
- While error types are categorized (Import Error, Missing Test, etc.), deeper causal insights (why import errors dominate, or which repository structures cause more failures) would make the analysis stronger.
- Insufficient documentation of evaluation details. Critical hyperparameters, such as temperature settings and the number of repeated runs per task, are not specified.
- The evaluation uses only 25 test instances per codebase, which is insufficient for drawing robust conclusions.
- The benchmark exclusively evaluates Python codebases, ignoring other programming languages (e.g., Java). This narrow focus limits insights into cross-language performance and fails to address challenges unique to diverse paradigms.

**Questions:**

- Could partial credit be integrated (e.g., per-test-case success rate) to provide smoother performance gradients?
- According to the definitions of Line Execution Rate, this metric is 100% when every line of the gistified file is executed. However, this may misalign with practical minimality when syntactically necessary code (e.g., unused branches in conditional statements) is retained for correctness but not executed under test inputs. For example, an if-else block where only one branch runs would penalize the metric score despite the code being functionally minimal.
- Could the line existence rate penalize valid simplifications?

---

> ### Author Response · Authors · 2025-11-20
>
> Hello Reviewer WQmg,
>
> We thank the reviewer for the time spent in reading and evaluating our submission and their valuable feedback.
>
> Here are our responses to the questions:
>
> > W1. The Execution Fidelity can only be signed to 1 or 0, which lacks granularity in capturing partially correct implementations.
>
> > Q1. Could partial credit be integrated (e.g., per-test-case success rate) to provide smoother performance gradients?
>
> Our current framework supports a more granular execution feedback; given that our tests are parametric and accept input-output pairs, we can report the average, rather than the floor performance across all I/O pairs. However, we believe that measuring success as satisfying all input-output pairs to be a cleaner performance metric. This avoids inflating the performance of a model which can only solve most of the easier entries, while struggling on the few harder ones.
>
> > W2. While error types are categorized (Import Error, Missing Test, etc.), deeper causal insights (why import errors dominate, or which repository structures cause more failures) would make the analysis stronger.
>
> To gain deeper causal insight into the errors, we analyzed the execution trajectories to identify interesting common patterns in model behavior. In particular, we perform a detailed analysis over GPT-5 and Claude-4 using the SWE-Agent framework.
>
> First, we observe that “missing test function” errors occur more as the model waits before creating the gistified file. For GPT-5, missing test functions tend to occur when the model creates the file later in the trajectory (avg. 88.0% of the trace), whereas traces without missing test function errors show earlier file creation (avg. 73.5%). Claude-4, in contrast, tends to create the file earlier (avg. 64.5%), which likely contributes to its lower rate of missing test function errors (3 instances for Claude-4 vs 72 for GPT-5). Also, we observe that Claude-4 performs more iterations on the gistified file, an average of 4.0 create or revision actions per test case, whereas GPT-5 makes actions only about 1.4 times. This suggests that creating the file early and iterating on it would be a more effective strategy.
>
> Second, when examining which files the models view across different test functions from the same codebase, we observe that Claude-4 accesses more diverse files, whereas GPT-5 tends to view similar files regardless of the test. On average, Claude-4 touches 41% more unique files than GPT-5 across all repositories. This broader coverage likely helps Claude-4 view and incorporate the correct, test-specific files, contributing to its higher overall performance. However, accessing and integrating multiple files can also introduce runtime errors, such as syntax errors.
>
> We also performed a repository-level analysis but did not observe any clear trends. However, our test-level analysis in Section 5.2 shows that as the length of the trace and the number of unique files increase, the model tends to find the task more difficult.
>
> > W3. Insufficient documentation of evaluation details. Critical hyperparameters, such as temperature settings and the number of repeated runs per task, are not specified.
>
> Thank you for pointing this out. We provide clarifications here. All experiments were executed with the same base set of hyperparameters. We use a default temperature of 1, a context window of 128K, and perform a single run for each configuration. The prompts we used are available in the appendix of the paper. For framework-specific hyperparameters (e.g. cache management, system prompts, tools), we use the default configuration made available by each framework, except for vscode-copilot-chat, where we have listed in the appendix the tools used in the experiments. We have added these details in the appendix.
> To summarize, the numbers in Table 1 are an average of 6x25=150 test examples. For each configuration of (framework, model, use execution tools) we ran 25 tests from each of the six repositories, each test executed only once.
> We are happy to discuss any other additional experimental design choices. We plan on releasing the code upon publication to facilitate the reproduction of our results.
>
> > W4. The evaluation uses only 25 test instances per codebase, which is insufficient for drawing robust conclusions.
>
> We provide additional results on a larger test set (2.5k) in general comments, showing that they align well with our original findings. For clarification, we would like to emphasize that the choice of 25 tests per repository is due to API cost and latency rather than the number of tests that GISTIFY can generate from the repository.

---

> ### Author Response · Authors · 2025-11-20
>
> > W5. The benchmark exclusively evaluates Python codebases, ignoring other programming languages (e.g., Java). This narrow focus limits insights into cross-language performance and fails to address challenges unique to diverse paradigms.
>
> We believe that the GISTIFY procedure of extracting a specific functionality of a given codebase can be adapted to arbitrary languages. For compiled languages with strict filename restrictions (Java, C#), we can soften the single-file constraint and keep the file structure as-is, but prune any unused code or files.
> Our contribution lies in the procedural way to generate tests from a given codebase, rather than the specific tests used in the paper, and we believe this procedure can be adapted beyond Python.
>
> > Q2. According to the definitions of Line Execution Rate, this metric is 100% when every line of the gistified file is executed. However, this may misalign with practical minimality when syntactically necessary code (e.g., unused branches in conditional statements) is retained for correctness but not executed under test inputs. For example, an if-else block where only one branch runs would penalize the metric score despite the code being functionally minimal.
>
> To prevent misalignment between minimality and line execution rate, we compute the Line Execution Rate by considering only executable and import lines, while excluding control-flow and definition lines.
> Control-flow and definition lines are excluded because their execution is often input-dependent; in some cases, the metric’s execution fidelity may be influenced by such edge cases by choice of parameter, and certain lines (e.g., else statement) may never execute but are required for correct syntax.
> For example, if the original code is:
> ```
> If x<0:
>    b() # this line ends up never being executed
> Else:
>    c() # this line is being executed
> ```
> And the gistified file is:
> ```
> c() # we do not penalize
> ```
> ```
> If x<0:
>    b() # this is penalized as it is executable
> Else:
>    c()
> ```
> ```
> If x<0:
>    Pass # this is not penalized as it is a placeholder
> Else:
>    c()
> ```
> We wrote a more detailed explanation in Appendix B.1 under “Normalization for Line-Wise Code Matching.” We could expand on executable or import lines for calculating line execution, which will likely require an additional module (e.g., LLM-based judge) to determine.
>
> Additionally, during the rebuttal, we conduct a user study to evaluate whether the Line Execution Rate reflects practical minimality. We observed an average Cohen’s kappa score of 0.61, indicating substantial agreement between the metric and users’ perception of ‘minimal’ code. More details on the user study are provided in Appendix F of the updated PDF.
>
> > Q3. Could the line existence rate penalize valid simplifications?
>
> We have added line normalization (e.g., indentation, multi-line) and a block-matching (checking lines by block) procedure so that the line existence rate does not penalize valid simplification. We have written a detailed algorithm in Figure 5 of the updated PDF.

---

> > ### Comment · Reviewer_WQmg · 2025-11-27
> >
> > Thanks for the response, I will keep my score.

---

> > > ### Author Response · Authors · 2025-11-27
> > >
> > > Thank you for taking the time to review our rebuttal. We would greatly appreciate knowing whether our response has addressed the reviewer’s concerns. Are there remaining concerns that you would like to see us address? Do you have any suggestions to make the paper stronger in the future?

---

> > > > ### Comment · Reviewer_WQmg · 2025-11-28
> > > >
> > > > Thanks for following up on my decision.
> > > >
> > > > There are some points that have not been fully addressed:
> > > >
> > > > - The rationale for the evaluation choices discussed in W1/Q1 remains unclear without empirical evidences.
> > > > - The results under temperature=1 are generally not quite deterministic. Any evidences indicating the relieability would help.
> > > > - The discussion in Q2 briefly mentions the potential use of an LLM-based judge module to improve the Line Execution metric, but the paper does not present evidence evaluating this addition.

---

> > > > > ### Author Response · Authors · 2025-12-01
> > > > >
> > > > > Thank you for your follow-up questions. Our responses are as follows:
> > > > >
> > > > > > The rationale for the evaluation choices discussed in W1/Q1 remains unclear without empirical evidences.
> > > > >
> > > > > As requested by the reviewer, we experimented with partial scoring based on success rates per parameter and observed that the results were **identical** to binary evaluation. When the model fails on one parameter, it typically fails on all, and similarly for success, likely because missing or incorrectly copying a function affects all inputs. Along with this, we argue that binary evaluation is more appropriate for Gistify, whose goal is to generate a self-contained file that **fully replicates the original behavior**. Partial credit could inflate scores even when the file does not operate correctly, and it assumes all parameter cases are equally important, which is not always valid.
> > > > >
> > > > > > The results under temperature=1 are generally not quite deterministic. Any evidences indicating the relieability would help.
> > > > >
> > > > > While temperature = 1 introduces some non-determinism, our evaluation reports performance **averaged over many points**, which ensures reliable and stable results. Additional experiments included in the rebuttal further confirm this stability. We would also like to highlight that temperature = 1 was chosen because **it is the default for both Claude and GPT-5**; GPT-5 no longer exposes a temperature argument and uses this default internally.
> > > > >
> > > > > > The discussion in Q2 briefly mentions the potential use of an LLM-based judge module to improve the Line Execution metric, but the paper does not present evidence evaluating this addition.
> > > > >
> > > > > As requested by the reviewer, we experimented with **adding an LLM-as-Judge module to the Line Execution metric**, allowing us to consider not only “executable and import lines” but also “control-flow and definition lines” (e.g., try:, else:), which were skipped in the standard metric because they exist as structural lines rather than lines actually executed. Given the tracing output for each line, the **LLM determined whether these lines would be executed according to Python semantics and runtime control flow**: lines are marked “executed” if the interpreter enters them along any execution path, and “not executed” if they cannot be reached. Following [1], we used Gemini-2.5-pro as the Judge model, as it is well-suited for this task in the code domain.
> > > > >
> > > > > We re-evaluated the results using the SWE-Bench framework with the LLM-as-Judge module added. We observed that including these additional lines slightly decreases the line execution rate, which we attribute to rarely executed lines, such as except:, being taken into account. The largest drop observed was 1.3% (GPT-5-mini). Overall, however, **the relative trends across models remain consistent**. Further analysis shows that control-flow and definition lines make up approximately 4.1% of lines in a gistified file, suggesting why adding the LLM-based judge has minimal impact on the metric.
> > > > >
> > > > > | Line Execution | original evaluation | with LLM-as-Judge   |
> > > > > |------------|-----|--------------|
> > > > > | GPT-5-mini     | 74.8 | 73.5 |
> > > > > | GPT-5        | 81.7  | 81.3 |
> > > > > | Claude-3.7    | 69.9  | 69.1  |
> > > > > | Claude-4    | 72.9  | 71.8 |
> > > > > [1] CodeJudgeBench: Benchmarking LLM-as-a-Judge for Coding Tasks

---

### Official Review · Reviewer_C2sw · 2025-10-30

**Soundness:** 3
**Presentation:** 3
**Contribution:** 3
**Rating:** 6
**Confidence:** 3

**Summary:**

This paper introduces **GISTIFY**, a new repository-level evaluation benchmark where LLM agents must compress a command's behavior (e.g., a specific pytest) into a single, self-contained file. The authors define clear task requirements, propose metrics for execution fidelity and minimality, and benchmark four leading LLMs across three agent frameworks on five SWE-Bench repositories plus debug-gym. The evaluation includes error analysis, ablations on prompting strategies and tools, and a difficulty analysis based on execution coverage.

**Strengths:**

- Unlike prior repository-level benchmarks (SWE-Bench, RepoBench), GISTIFY uniquely requires runtime-guided single-file reconstruction. This frames an interesting and practical challenge.
- The paper tests four models across three frameworks, includes thoughtful ablations on prompting strategies and tool usage, and extracts concrete insights about agent behavior and design trade-offs.
- The task is well-explained, metrics are intuitive, and the error categorization (import errors, missing functions, etc.) provides useful diagnostic insights.
- The finding that state-of-the-art agents still struggle on long execution traces is noteworthy and motivates future work on execution-aware reasoning.

**Weaknesses:**

- The main experiments cover only 25 tests per repository across 5 Python projects. It's unclear how well this generalizes to larger codebases, other languages, or more diverse domains. The repository selection criteria and expansion plans are not well explained.
- The line execution/existence rates are intuitive but not validated. Without human reference gists or user studies, it's hard to know if these metrics truly measure "minimal and faithful" distillations or just correlate with them.

**Questions:**

1. Do you have plans to expand the benchmark beyond 25 tests per repo? What's the timeline, and will the expanded sets be publicly available?
2. Have you considered validating the minimality metrics against human judgments? Even a small pilot study would strengthen confidence in your metrics.

---

> ### Author Response · Authors · 2025-11-20
>
> Hello Reviewer C2sw,
>
> We thank the reviewer for their feedback and the time taken to assess the correctness and relevance of the submission.
>
> Here are our responses to the questions:
>
> > W1. The main experiments cover only 25 tests per repository across 5 Python projects. It's unclear how well this generalizes to larger codebases, other languages, or more diverse domains. The repository selection criteria and expansion plans are not well explained.
>
> > Q1. Do you have plans to expand the benchmark beyond 25 tests per repo? What's the timeline, and will the expanded sets be publicly available?
>
> We provide additional results on a larger test set (2.5k) in the general comments, showing that they align well with our original findings. For clarification, we would like to emphasize that the choice of 25 tests per repository is due to API cost and latency rather than the number of tests that GISTIFY can generate from the repository.
>
> To ensure repository quality, we selected widely used GitHub repositories included in SWE-Bench. To further demonstrate that our GISTIFY framework generalizes beyond these repositories, we also conducted experiments on a repository not included in SWE-Bench (Section 4.1).
>
> Furthermore, we have already expanded the benchmark to over 2.5k tests and plan to release the code upon acceptance, enabling the community to easily extend the benchmark according to their needs.
>
> > W2. The line execution/existence rates are intuitive but not validated. Without human reference gists or user studies, it's hard to know if these metrics truly measure "minimal and faithful" distillations or just correlate with them.
>
> > Q2. Have you considered validating the minimality metrics against human judgments? Even a small pilot study would strengthen confidence in your metrics.
>
> To validate whether line-execution and line-existence metrics reflect minimality and faithfulness, we conduct a pilot study to validate these metrics against human judgment.
> We recruit three software/AI engineers and provided them with pairs of gisitifed files from the same test set, generated by different models or frameworks. Annotators were asked to evaluate which file better satisfied the definition of “minimal” and “faithful” (see Appendix F of the updated PDF for detailed instructions provided to annotators).
>
> The results show an average Cohen’s kappa score of 0.61 for minimality aligned with line-execution and 0.76 for faithfulness aligned with line-existence, indicating that the metrics correspond well with human judgements of minimality and faithfulness.

---

### Official Review · Reviewer_TALW · 2025-10-31

**Soundness:** 3
**Presentation:** 2
**Contribution:** 3
**Rating:** 6
**Confidence:** 3

**Summary:**

This paper introduces GISTIFY, a benchmark for evaluating codebase-level understanding in large language models. Given a repository and a command, the model is required to generate a minimal, self-contained file that reproduces the same runtime output as the original codebase. The benchmark is automatically constructed from real GitHub projects and paired with execution traces that capture ground-truth runtime behavior.Evaluation metrics include execution fidelity, line existence rate, and line execution rate. Experiments with multiple LLMs and agent frameworks show that while models can handle simple single-module tasks, they struggle on multi-file or long-trace scenarios.

**Strengths:**

This work introduces a new approach to evaluate models' code repair ability from a more comprehensive and systematic perspective. The benchmark is built from real execution traces, collected via an automated pipeline, and the evaluation criteria are well thought out.

**Weaknesses:**

1. Since code execution inevitably produces different errors due to various factors, it is unlikely that data filtering alone can fully prevent them. How should such errors be handled during evaluating? Could there be missing special filtering or handling mechanisms?
2. The benchmark is quite novel, but the overall size might be somewhat limited.

**Questions:**

See weakness please

---

> ### Author Response · Authors · 2025-11-20
>
> Hello Reviewer TALW,
>
> Thank you for your valuable time in reviewing our paper. We address below the points raised by the reviewer.
>
> > W1. Since code execution inevitably produces different errors due to various factors, it is unlikely that data filtering alone can fully prevent them. How should such errors be handled during evaluating? Could there be missing special filtering or handling mechanisms?
>
> We apply a filtering process before running the test to remove examples that may produce inevitable errors, such as tasks requiring network access or those containing hard-coded relative paths. Also, to further reduce environment-related issues (e.g., package version, file system restrictions), we ran the experiment using the same Docker image.
> Additionally, when analyzing the results reported in Table 2, we manually inspected a subset of cases and could see that the errors stem from model behavior (e.g., missing test functions, incorrect imports), rather than external or environment-related factors.
>
> > W2. The benchmark is quite novel, but the overall size might be somewhat limited.
>
> We provide additional results on a larger test set (2.5k), showing that they align well with our original findings in the general comments. For clarification, we would like to emphasize that the choice of 25 tests per repository is due to API cost and latency rather than the number of tests that GISTIFY can generate from the repository.

---

> > ### Comment · Reviewer_TALW · 2025-11-27
> >
> > Thank you for the clarification. I apologize for having overlooked some key information in the paper earlier. My concerns have been addressed, so I maintain my positive rating.

---

> > > ### Author Response · Authors · 2025-11-27
> > >
> > > Thank you for taking the time to review our rebuttal. We are pleased to hear that the concerns have been addressed and that you are recommending the paper for acceptance.

---

### Official Review · Reviewer_SdHh · 2025-11-01

**Soundness:** 3
**Presentation:** 3
**Contribution:** 4
**Rating:** 6
**Confidence:** 4

**Summary:**

GISTIFY tests whether LLMs truly understand whole codebases—not just snippets—by requiring them to generate a single minimal Python file that exactly replicates a command’s runtime behavior (e.g., pytest output). Success demands four criteria: files must be self-contained (inline dependencies), execution-faithful (identical output), minimal (only essential code), and grounded (zero hallucinations, strictly sourced). Evaluating top models (GPT-5, Claude-4) via agent frameworks like SWE-Agent, the authors introduce three practical metrics: binary execution fidelity, Line Execution Rate (minimality), and Line Existence Rate (source fidelity). Results reveal even leading models struggle significantly with complex tasks, especially long dependency chains. Error analysis shows failures stem from incorrect inlining or over-pruning critical code. This benchmark exposes core weaknesses in current LLMs' cross-file reasoning.

**Strengths:**

1. This paper addresses a critical gap in current LLM code evaluation, which leans heavily on static analysis. Requiring full runtime behavior replication forces models to grasp entire codebase execution flows—crucial as AI coding assistants scale toward industrial systems handling multi-file complexity.
2. With just a repository and entrypoint command, GISTIFY generates evaluation cases automatically for any codebase without relying on manually labeled data such as GitHub issues. This sidesteps annotation biases and enables direct cross-project comparisons.

**Weaknesses:**

1. The main results in Table 1 are based on 25 tests from each of 5 repositories. While the analysis section uses a larger set of 50 tests from pylint, the primary claims about model performance would be more robust if supported by a larger and more diverse set of test instances in the main experiment.

**Questions:**

1. How does the normalization and block-matching algorithm handle semantically equivalent but syntactically different code changes, such as rewriting a for-loop as a list comprehension or rearranging independent class methods? Could you include the relevant pseudocode or a more detailed description of the matching logic?
2. For handling non-deterministic program outputs (e.g., timestamps, memory addresses, or unordered dictionary iteration in legacy Python versions), what normalization methodology is applied to stdout/stderr prior to comparison?
3. In this experiment, the static model receives a strong oracle. The observed conclusion indicates dynamic file selection achieves superior performance. It is hypothesized that the static model's suboptimal results may stem from context overload. Could implementing alternative prompting strategies—such as single-turn multi-step Chain-of-Thought (CoT) that directs initial identification of essential code segments before file synthesis—effectively mitigate this performance gap?

---

> ### Author Response · Authors · 2025-11-20
>
> Hello Reviewer SdHh,
>
> We thank the reviewer for their valuable feedback and the time taken to assess our paper! Here are our responses to the questions:
>
> > W1. The main results in Table 1 are based on 25 tests from each of 5 repositories. While the analysis section uses a larger set of 50 tests from pylint, the primary claims about model performance would be more robust if supported by a larger and more diverse set of test instances in the main experiment.
>
> We provide additional results with a larger test set (2.5k), showing that they align well with our initial results in the general comments. For clarification, we would like to emphasize that the choice of 25 tests per repository is due to API cost and latency rather than the number of tests that GISTIFY can generate from the repository.
>
> > Q1. How does the normalization and block-matching algorithm handle semantically equivalent but syntactically different code changes, such as rewriting a for-loop as a list comprehension or rearranging independent class methods? Could you include the relevant pseudocode or a more detailed description of the matching logic?
>
> In addition to the details originally mentioned in Appendix (B.1), we have updated the PDF with detailed pseudo-code (Appendix Figures 3 and 4) and revised the paragraph titled “Normalization for Line-wise Code Matching” in Appendix B.1 to incorporate explanations regarding the added pseudo-code.
>
> When calculating line existence, our normalization and block-matching procedure handles simple syntactic reordering. For example, rearranging independent class methods does not affect matching because classes are represented as dictionaries and compared by key rather than by order. In contrast, as the purpose of our benchmark is to assess whether the model can understand and extract the runtime execution, we treat larger syntactic rewrites, such as replacing a for-loop with a list comprehension, as mismatches even if they are semantically equivalent. Given that the model is explicitly prompted not to rewrite or reformat lines unless necessary, we penalize accordingly.
>
> > Q2. For handling non-deterministic program outputs (e.g., timestamps, memory addresses, or unordered dictionary iteration in legacy Python versions), what normalization methodology is applied to stdout/stderr prior to comparison?
>
> Thank you for raising this point. Our current experiment focuses on deterministic cases where tests serve as the entry point, which avoids issues with non-deterministic outputs. We agree that such cases with non-deterministic program output may arise when extending to arbitrary entry points, which can be mitigated by filtering out test cases involving non-deterministic behavior or by using an LLM-based judge for output comparison. We added this discussion to the limitation section in the updated version. Importantly, we would like to emphasize that even in such cases, GISTIFY remains an easily extensible yet difficult benchmark.
>
> > Q3. In this experiment, the static model receives a strong oracle. The observed conclusion indicates dynamic file selection achieves superior performance. It is hypothesized that the static model's suboptimal results may stem from context overload. Could implementing alternative prompting strategies—such as single-turn multi-step Chain-of-Thought (CoT) that directs initial identification of essential code segments before file synthesis—effectively mitigate this performance gap?
>
> Thank you for the suggestion. Following the reviewer’s suggestion, we ran a new variant of the oracle static model with additional single-turn multi-step CoT prompting, requiring the model to first identify essential code segments before generating the final file. We have updated the PDF with the result (Figure 20), prompt (Figure 21), and details in Appendix E.2.
>
> Even with this additional CoT prompting, we observe the same overall trend as in the baseline static model: although the CoT-augmented static model shows a small performance gain (+3.6), it still underperforms the dynamic models (avg -5.9), underscoring the importance of multi-turn generation and tool usage to perform the GISTIFY task effectively.

---

> > ### Comment · Reviewer_SdHh · 2025-11-27
> >
> > I thank the authors for their detailed and thoughtful response. The rebuttal has effectively addressed my main concerns.
> > I recognize that perfect code normalization is a difficult problem, and while there is room for improvement, the proposed method is reasonable. The paper is a good contribution, and I maintain my recommendation to accept.

---

> > > ### Author Response · Authors · 2025-11-27
> > >
> > > Thank you for taking the time to review our rebuttal. We are pleased to hear that the concerns have been addressed and that you are recommending the paper for acceptance.

---

### Author Response · Authors · 2025-11-20

Dear Reviewers, we thank you for your valuable feedback! We really appreciate the time you spent providing us with constructive advice to improve our submission. Below, we list our general response to all of you, and address the individual concerns in separate threads.

Also, we have updated the PDF with revisions highlighted in red. The main updates are as follows:
- Figure 3-5: added pseudo code for line execution and existence metric
- Appendix E.2: experiment over static coding LLM with single-turn, multi-step chain-of-thought prompt
- Appendix F: a user study analyzing the correlation between line execution/existence and the definitions of minimality and faithfulness.


## General Comment: On the size of the benchmark

We wish to clarify that **Gistify! is a procedural way to generate evaluation samples. It is not a fixed dataset, but rather an approach to automatically generate testing examples.** For example, across the 6 codebases considered in this paper, we have over 40K tests on which to evaluate the gistify task. Also, we can easily expand to any new codebase using the same procedure when an entry point and original codebase are given.

That being said, we understand the reviewers' concern that our main analysis should contain more examples; we have limited our results to 150 samples per configuration, given the considerable API cost of frontier models. Please note that we have conducted experiments over three different frameworks and four different models, along with ablation studies over different tools and prompt usage. To motivate our experimental choice, we ran the following configuration on **16.7x** more samples from the 6 repositories we experimented upon using GPT-5-mini with the SWE-Agent framework, and we show that **these extended results align with our initial numbers**:

| # of Test Instances | Execution Fidelity (w/ exec) | Line Existence | Line Execution |
|----------|----------|----------|----------|
|   150  | 45.3     |47.9     | 74.8 |
| 2.5k   | 46.4     | 47.8     | 75.6|

We therefore conclude that our initial sample size provides a meaningful estimate of the performance for a given configuration. We are also running additional experiments on GPT-5, which require longer API calls, and will share the results once they are available.

---

### Author Response · Authors · 2025-12-03
**Summary for Area Chair**

This note summarizes the strengths highlighted by reviewers and the remaining ongoing discussions before the cutoff caused by the OpenReview bug.

## Strength
- **Novelty and significance of the task**: All reviewers agreed the task is important, filling a key gap in LLM code evaluation by requiring full runtime and execution-flow understanding (SdHh, WQmg, C2sw, TALW). They also highlighted its automatic applicability to any codebase (SdHh, TALW) and its difficulty even for state-of-the-art agents (C2sw).
- **Strong metric design**: The reviewers mentioned that the suggested metrics provide a comprehensive evaluation of the task (WQmg), are well thought out (TALW), and are intuitive (C2sw).
- **Well-designed experiments and analysis**: The reviewers mentioned that experiments offer comparative insights across multiple models and frameworks (WQmg), ablations on prompting and tool usage share concrete insights (C2sw).

## Weakness & Questions:
**Reviewer SdHh and TALW respond to the rebuttal that their concerns are addressed and maintain their recommendation to accept.** Below, we **summarize remaining discussions that were cut off early** due to an OpenReview bug.

### Reviewer C2sw:
> Q1. Limited number of test cases
- We clarified that **Gistify! is a procedural method for automatically generating evaluation samples, not a fixed dataset**.
- To validate our results, during rebuttal, **we experimented with over 16.7× more samples** and observed **trends consistent** with our initial findings.

> Q2. User studies on whether line execution and existence align with human judgment
* During rebuttal, a user study with three software/AI engineers showed that **“minimal” and “faithful” definitions align well with line execution and existence** (Cohen’s kappa: 0.61 and 0.76, respectively). See Appendix F in the updated PDF for details.

### Reviewer WQmg:
We had a discussion and received three remaining questions, but the conversation was cut off by the OpenReview bug.

> W1, Q1: Rationale of binary score remains unclear without empirical evidence
* We conducted additional experiments with partial scoring and found results **identical** to the binary metric, as missing or incorrectly copying a function affects all inputs.

> W3: results under temperature=1 are generally not quite deterministic
* We clarified that temperature 1 is the **default setup** for Claude and GPT-5 (temperature is fixed for GPT-5) and that averaging over many evaluation points ensures reliable and stable results.

> Q2: Using an LLM-based judge for control-flow and definition lines in the Line Execution metric
* We tested with the LLM-as-judge module added for these lines; **relative model trends remained consistent**, with minimal impact, as control-flow and definition lines comprise only ~4.1% of a gistified file.

---

### Meta-Review · Area_Chair_G9BT · 2026-01-06

**Summary:**

The following five types of observations can be summarized below:
1. Methodological Novelty. Reviewers universally acknowledged the significance of the Gistify task. Unlike previous benchmarks relying on static analysis, this work requires agents to understand runtime execution flow to generate self-contained, minimal, and faithful code. The proposed "runtime execution" paradigm is seen as a critical step forward for coding agent evaluation.

2. Sample Size and Robustness. A primary initial concern was the limited sample size (25 tests per repository). However, the authors clarified that Gistify is a procedural method rather than a fixed dataset. To demonstrate robustness, they expanded the test set to 2,500 instances during the rebuttal, showing that performance trends remained consistent (e.g., Execution Fidelity 46.4% vs 45.3%).

3. Metric Validity. The introduction of new metrics (Line Execution/Existence Rate) raised questions regarding their alignment with human perception of "minimality" and "faithfulness". The authors addressed this via a user study, yielding Cohen’s kappa scores of 0.61 and 0.76. These scores indicate moderate agreement, suggesting that while the metrics are useful proxies, they are not perfect substitutes for human judgment.

4. Experimental Setup and Efficiency. The paper provides comprehensive comparisons across frameworks (SWE-Agent, Copilot) and models (GPT-5, Claude-4). While the findings on long execution traces and agentic workflows are valuable, the argument for cost efficiency remains weak. Evaluating a model using GPT-5 driven SWE-agent is resource-intensive and hardly "lightweight," and the authors failed to provide direct comparisons of wall-clock time or cost against standard RAG or fine-tuning pipelines.

5. Generalization and Complexity. The claim regarding generalization beyond Python and unit test commands remains unproven. Furthermore, performance drops significantly (to ~21%) on the constructed "GISTIFY-hard" subset, indicating that the method's ability to generalize degrades sharply as task complexity increases, even within its own narrow domain.

**Reviewer Concerns:**

The reviewer's concerns may be addressed by the rebuttal:
1. Sample Size: The authors expanded the experiment from 150 to 2,500 test instances, confirming the reliability of the initial findings.
2. Metric Validity: The authors conducted a human user study showing substantial agreement (Kappa 0.61/0.76) between the proposed metrics and human judgment.
3. Static Baseline Weakness: The authors added a single-turn multi-step Chain-of-Thought (CoT) baseline, which still underperformed dynamic agents, reinforcing the need for agentic workflows.
4. Handling Valid Unexecuted Code: The authors clarified that control-flow lines are excluded from the minimality metric and validated this by adding an LLM-as-a-Judge module, which resulted in negligible performance shifts (<1.3%).

The reviewer's concerns may still be outstanding:
1. Granularity of Binary Scoring: Reviewer WQmg maintained that a binary (0/1) score for Execution Fidelity lacks granularity. While the authors provided empirical evidence that partial scoring yields identical results, the qualitative justification remains a point of contention.
2. Determinism of Temperature=1: Reviewer WQmg remained skeptical about the reliability of results generated with temperature=1 , despite the authors' explanation that stability is achieved via averaging.
3. Cost and Efficiency: The lack of a rigorous cost analysis (vs. RAG or fine-tuning) leaves the claim of Gistify being a "lightweight" benchmark open to debate.

**Reviewer Scores:**

1. For Reviewer SdHh, they maintained their score (6), because the rebuttal effectively addressed the concerns regarding code normalization and matching logic.
2. For Reviewer TALW, they maintained their score (6), because the concerns regarding sample size and error handling were resolved by the additional data and filtering explanations.
3. For Reviewer C2sw, they maintained their score (6), because the user study provided the necessary validation for the proposed metrics.
4. For Reviewer WQmg, they may not change their score (4), because they retain reservations regarding the lack of empirical evidence for the reliability of Temperature=1 and the binary nature of the scoring metric.

---

### Decision · Program_Chairs · 2026-01-26

Accept (Poster)